# Mathematical Modeling for the Assessment of Public Policies in the Cancer Health-Care System Implemented for the Colombian Case

**DOI:** 10.3390/ijerph20186740

**Published:** 2023-09-11

**Authors:** Daniel Rojas-Díaz, María Eugenia Puerta-Yepes, Daniel Medina-Gaspar, Jesús Alonso Botero, Anwar Rodríguez, Norberto Rojas

**Affiliations:** 1Area of Fundamental Sciences, School of Applied Sciences and Engineering, Universidad EAFIT, Medellin 050022, Colombia; 2School of Finance, Economics, and Government, Universidad EAFIT, Medellin 050022, Colombia; 3Center for Economic Studies, National Association of Financial Institutions (ANIF), Bogota 110231, Colombia

**Keywords:** cancer care, discrete time, identifiability analysis, health system, mathematical modeling, parameter estimation, public health, public policies, sensitivity analyses

## Abstract

The incidence of cancer has been constantly growing worldwide, placing pressure on health systems and increasing the costs associated with the treatment of cancer. In particular, low- and middle-income countries are expected to face serious challenges related to caring for the majority of the world’s new cancer cases in the next 10 years. In this study, we propose a mathematical model that allows for the simulation of different strategies focused on public policies by combining spending and epidemiological indicators. In this way, strategies aimed at efficient spending management with better epidemiological indicators can be determined. For validation and calibration of the model, we use data from Colombia—which, according to the World Bank, is an upper-middle-income country. The results of the simulations using the proposed model, calibrated and validated for Colombia, indicate that the most effective strategy for reducing mortality and financial burden consists of a combination of early detection and greater efficiency of treatment in the early stages of cancer. This approach is found to present a 38% reduction in mortality rate and a 20% reduction in costs (% GDP) when compared to the baseline scenario. Hence, Colombia should prioritize comprehensive care models that focus on patient-centered care, prevention, and early detection.

## 1. Introduction

Both the costs associated with cancer healthcare and the incidence rate of cancer are on the rise worldwide. It has been estimated that the global cancer cost, as a percentage of GDP, reached 0.47% in the year 2018 [1]. Furthermore, without changes in current health systems, the world could expect 76 million cancer deaths between 2020–2030, mainly concentrated in lower- to middle-income populations (70%) [1]. However, Ward et al. [1] have also suggested that the comprehensive scaling-up of treatment, imaging, and quality of care could avert 5–12% of these deaths globally, producing USD (2018) 2.9 trillion in lifetime economic benefits while costing an additional USD 232.9 billion between 2020 and 2030 (associated with a 6–9% increase in cancer treatment costs), yielding a return of USD 12.43 per USD 1 invested. As such, countries should strive to develop health system configurations with better health outcomes, such as lower mortality and incidence rates and improved quality of life for patients, while also being financially sustainable in the long term [2]. Thus, the implementation of strategies that improve health systems must include a cost-effectiveness analysis [3,4].

Mathematical modeling applied to cancer—from the gene to population level—has drawn significant attention over the last two decades, due to its potential to help decision-makers in discerning among different strategies to improve certain components of the cancer healthcare system [5,6]. Mathematical models can capture more of the dynamics of cancer disease by integrating the increasing volume of available data and simulating the interactions between the actors involved in cancer healthcare [7]. However, most of the research so far has mainly focused on cancer treatment, early detection, and prevention [7,8,9]; which, in turn, has led to a lack of models treating the cancer healthcare system as a whole, despite the implementation of precision public health being a central issue in this field [10,11].

Due to the high complexity of both cancer dynamics and healthcare system dynamics, a remarkable amount of discrete-time approaches for mathematical modeling have been proposed in this area [7,12], especially regarding healthcare systems [13]. Another noticeable technique used to propose models in these areas from a holistic point of view is the system dynamics approach [14]. Both approaches can be applied at the same time, in order to achieve a holistic and robust framework that takes into account the complexity of the different components that constitute the whole system [12]. This is especially useful for precision public health, as any public health policy must be both viable in terms of health outcomes and financially sustainable, given that public resources are limited and access to healthcare must be guaranteed for all [2,15]. Specifically for cancer, available strategies are challenging on both fronts due to various factors including high mortality rate, costly treatment with high uncertainty, and supply restrictions for specialized personnel, medical equipment, and access to technologies, as well as difficulties related to early detection [7,9].

In this paper, inspired by the work of Catano-Lopez et al. [16], we propose a strategy to model cancer healthcare systems. Most of the information available is included in a discrete-time structure motivated by the natural history of the disease, and we also allow for the assessment of public health policies by taking the Colombian health system as a case study. As suggested by Davahli et al. [15], we focus our attention on the flow of patients inside the healthcare system. As public policies for cancer treatment must balance health outcomes and financial sustainability to make them viable in the long term [1], we adapt the model to the actual state of the Colombian cancer healthcare system, which reveals some interesting dynamics, following which we use the model to test several hypotheses on cancer healthcare strategies. This allows us to to look for scenarios characterized by moderate spending and improved long-term health outcomes of the system. In other words, we assess a wide combination of scenarios and test their capacity to control the spending curve while improving population health outcomes.

## 2. Materials and Methods

### 2.1. Modeling Approach

We follow the guidelines stated by Martcheva [17] to propose a discrete-time compartmental mathematical model. In this way, the developed model can be treated as a graph, where the flow among nodes is described through probabilities or proportions [16]. The intuition behind compartmental mathematical models is to propose a partition of the population under study according to some properties observed in that population; then, each component of the partition becomes a compartment (also known as a state) [17]. The subpopulation within a compartment may vary over time, according to the flows and interactions established between compartments; for instance, let $\left\{X_{1}\left(t\right),\cdots ,X_{m}\left(t\right)\right\}$, t∈{0,1,⋯,k} represent a partition of some population of interest as time evolves (in a discrete manner) and $X_{i}\left(t\right)$, i∈{1,⋯,m} be the subpopulation exhibiting some particular conditions. Then, an equation describing the evolution of the subpopulation in this state would be as follows: $$X_{i}\left(t+1\right)=I_{i}\left(X_{1}\left(t\right),\cdots ,X_{m}\left(t\right),t\right)+X_{i}\left(t\right)O_{i}\left(X_{1}\left(t\right),\cdots ,X_{m}\left(t\right),t\right),$$
where $I_{i}\left(\cdot \right)$ is a function of the amount of the population in state *i* (linked to $X_{i}\left(t\right)$), as well as the time, acting as the inflow of population to the ith state for the next time step. Similarly, $O_{i}\left(\cdot \right)$ is a function of the amount of population and the time that determines the proportion of the population in $X_{i}\left(t\right)$ that will not flow out in the next time step. In their simpler forms, $O_{i}\left(\cdot \right)$ could be a constant while $I_{i}\left(\cdot \right)$ would be a linear combination of outflows alongside a constant. For example, after setting $O_{i}\left(\cdot \right)=K_{i}$, we obtain the following representation: (1)$$X_{i}\left(t+1\right)=K_{0}+\sum_{j=1}^{m}K_{j}X_{j}\left(t\right).$$

Furthermore, for the modeling approach, we chose every parameter involved in the structure of the model to have a natural, useful, and intuitive meaning, while keeping the model free of dimensional issues. In this way, we can ensure that all of the parameters we inferred or estimated for the model can be easily contrasted with real data, even if they are unavailable at present, as they can be obtained in the future in simple studies.

### 2.2. Data Sources

To propose the model and conduct subsequent calibration of the values for some of its parameters, we used information available for Colombia in recent years from three major databases: the first one was Cuenta de Alto Costo (CAC), which is a government-independent organization belonging to the general social security system of the country, whose purpose is to stabilize the health system with regard to high-cost diseases, such as cancer [18]; the second one was SISPRO [19], a governmental database in which raw information about procedures in the Colombian health system is deposited; and the third one was Cancer Today, which is a web page that enables comprehensive assessment of the cancer burden worldwide in 2020, based on the Global Cancer Observatory (GLOBOCAN) estimates of incidence, mortality, and prevalence for year 2020 in 185 countries or territories for 36 cancer types by sex and age group [20]. Additionally, we used information from the National Administrative Department of Statistics DANE (its acronym in Spanish) regarding the population in the country and its decomposition into age groups from recent years up to the year 2070 [21].

Unlike the case of CAC and Cancer Today, the data stored in SISPRO are not publicly available, and prior access authorization is required. Additionally, it was necessary to download and curate the data from this source due to the presence of noticeable outliers. Considering the size of the SISPRO database, we only downloaded multivariate data for procedures related to 15 different types of cancer prioritized by the Colombian government. The dataset included the following variables: purpose of the procedure, procedure classification, type of health administrator for the patient, age group of the patient, municipality of residence of the patient, year in which the procedure was performed, cost of the procedure, and number of patients. It is important to note that most of the variables were categorical. For data downloaded from SISPRO, we utilized the software R along with the olapR package. The data curation process was performed using MATLAB. For further information about the authorization required, the data curation process, and the types of prioritized cancer, please refer to Appendix A.

### 2.3. Cancer Care in the Colombian Healthcare System

According to recent reports, the healthcare system in Colombia is a complex network that comprises financing, governance and organization, resource management, health service delivery, and evaluation and monitoring processes [22,23]. Private and public stakeholders participate in each component of the system. The Ministry of Health and Social Protection assumes most of the stewardship responsibility at the macro level, followed by local governments, which focus on providing care to specific populations. The Administrator of the Resources of the General Social Security Health System (ADRES) manages resources and performs the corresponding controls. Health insurance agencies (EPS) and providers are responsible for utilizing resources and delivering health services at the meso level. As pointed out in [23], Colombia operates within a managed competition model, in which individuals choose an EPS and access an IPS within their EPS network. At the meso and micro levels, entities can be found that perform specific functions in the system, such as the National Institute for Drug and Food Surveillance (Invima), the National Health Institute, the Agency of Health Technology Assessment (IETS), and the National Health Superintendence. The National Health Superintendence monitors health actors and imposes sanctions on those who do not comply with the regulations.

The healthcare system is financed through worker and employer contributions, general tax resources, private health expenditures, and donations [22]. It operates with a negative list, meaning that it finances most treatments, with exceptions mainly related to aesthetic issues and certain technologies not included in the basic health plan (PBS) [24]. According to [25], Colombia has one of the systems that best protects the population’s individual wealth.

Within this system, together with other actors that fulfill specific functions related to cancer (e.g., the National Cancer Institute, high-cost account, National Cancer Observatory, among others), entities define the ecosystem for the management of this disease. This works mainly under the rules established in the law Sandra Ceballos (Law 1384 of 2010), which establishes actions for the comprehensive care of cancer in Colombia, and its modification in Law 2194 of 2022. A timeline of key cancer policies can be found in [26].

In the above framework, a recent study on Colombia [27] provided new data regarding real access to high-quality diagnostic, curative, and palliative care for prioritized cancers in the national policy. The study reported a poor prognosis compared to high-income countries. Specifically, despite having achieved almost universal health coverage, Colombia still faces significant challenges in access to preventive, diagnostic, and treatment services for cancer patients. People living in poverty have lower access to all types of care, and other challenges due to a lack of health literacy, beliefs, and knowledge can be observed.

### 2.4. Natural History of the Disease

According to the Clínic Barcelona of the Universitat de Barcelona, the natural history of cancer has seven phases [28]. In Colombia, the Ministry of Health and Social Protection has proposed a similar disease history for breast cancer [29]. This starts with the preclinical period; here, secondary prevention through screening is effective in reducing incidence and mortality during this silent development phase. In the symptomatic phase, early detection is critical for timely diagnosis and treatment. An early diagnosis can lead to increased survival with adjuvant therapies and surgery, or improved survival and quality of life through advanced therapies. Following diagnosis, patients enter the control phase, where complications may occur, reducing their chances of survival. Ultimately, the patient may have two outcomes: long-term survival or palliative care.

In Colombia, CAC prioritizes 11 types of cancer, based on the tumor, lymph node, and metastasis (TNM) classification system (based on the relevance of disease burden and financial sustainability of the healthcare system, Resolution 3974 of 21 October 2009 prioritizes the following: breast cancer, cervical cancer, prostate cancer, colorectal cancer, stomach cancer, lung cancer, melanoma, Hodgkin’s lymphoma, non-Hodgkin’s lymphoma, acute lymphoid leukemia, and acute myeloid leukemia). The TNM system places cancer stages into five categories: stage 0 (in situ); stages I, II, and III (which indicate the presence of cancer); and stage IV (which indicates metastasis). In a characterization of the disease in the adult population made by CAC with data up to 2021, the authors showed a growing prevalence of cancer (at an annual rate of 14.7%) in recent years (393.6 in 2016 to 783.2 in 2021 per 100 thousand inhabitants), while the mortality rate increased at a rate of 4% per year in the same period [18]. Additionally, in the cases that were reported, a large proportion (48%) was identified in the advanced stages of the disease (i.e., III and IV), in which the highest mortality was concentrated [18], as well as the highest treatment costs. In fact, a study of the direct costs of breast cancer in Colombia found that moving from stage II to III increased the cost by 16%, while moving to stage IV did so by 125% [30].

In the same CAC study [18], a distinction between the different types of cancer revealed that, in 2021, 4.65% (1839) of new cancer cases in adults were in situ, while 95.35% (37,706) were invasive. Among the invasive cases, 90% were tumors and 5% were lymphomas. Approximately 65% of these cases were staged, with 8.7% being in situ, 18.7% stage I, 24.7% stage II, 23.6% stage III, and 24.1% stage IV. In terms of treatment, surgery was the most common initial management for those who received treatment (37.3% of the 39,323 new cases), with 98% receiving it as the initial treatment. Systemic therapy was administered as the initial curative treatment without surgery for 27.66% of cases, and as adjuvant therapy for 27.34% of cases. Radiotherapy was primarily used as adjuvant therapy (38.26% of cases), accounting for 15.13% of total treatments. At the cutoff date, 32.16% of new cases (12,716) had not received any treatment.

### 2.5. Fitting and Validation

We obtained short time-series for mortality, incidence, prevalence, and spending on cancer from the aforementioned sources; however, the nature of the data alongside the high number of parameters that we identified as necessary for the model led us to propose a model whose components could be separated from the whole structure, in order to deal with them under controlled scenarios and ensure the identifiability of the parameters. In this way, we performed parameter estimation routines for each component alone, then proceeded to assemble the calibrated components into the macro-structure. We also took mortality and spending as control variables, which means that we assessed the performance of the model by contrasting its outputs for these variables with the real data, instead of performing any fitting procedure for them.

To propose the final model as an assembling of components we chose the same approach of [16] to implement diffusion processes through defining some diffusion matrices. We implemented the model in MATLAB using the GSUA_CSB Toolbox, freely available at MathWorks file exchange [31], following the guide for user-defined models. The GSUA_CSB toolbox also allowed us to perform parameter estimation by minimizing the mean-squared error loss function via the MATLAB interior-point algorithm implemented in fmincon, and practical identifiability analysis following the guidelines stated by Lizarralde-Bejarano et al. [32]. Thus, we validated the model by checking out its goodness of fit regarding the control variables and assuring the estimated parameters to be locally identifiable. Briefly, a parameter is said to be locally identifiable when its output (Y), that depends on the time (*t*), all the current states of the system (X(t)), and a set of parameters (θ), given in the form
(2)$$Y\left(t\right)=g\left(t,X\left(t\right),\theta \right),\;\;\;x\left(0\right)=x_{0},$$
satisfies the following definition (taken from [33]):

**Definition** **1.***A system structure* (Equation 2) *is said to be locally identifiable if, for any **θ** within an open neighborhood of some point $\theta^{*}$ in the parameter space, $g\left(x\left(t\right),\theta_{1}\right)=g\left(x\left(t\right),\theta_{2}\right)$ holds if and only if $\theta_{1}=\theta_{2}$.*

However, checking Definition 1 in practice has been found to be very challenging [34]. Therefore, we tested whether the estimated parameters of our model held in the practical identifiability approach proposed by Lizarralde-Bejarano et al. [32]. For this approach, it suffices to focus on the dispersion of the estimations, as it is possible to suspect a parameter to be practically identifiable regarding the extent to which its dispersion tends to a single point. The higher the dispersion for a given parameter, the lower its identifiability. It is fundamental to check whether a parameter is identifiable, as they are the only values that can be reliably estimated.

To achieve intervals for estimates of the parameters, we decided to filter the estimates to keep only those which best fit the data; that is, the top 10–20% with the lowest loss values. To the ensure that the parameters are identifiable, it follows that those parameter values with best fit should also belong the same global minimum of the optimized loss function. Then, it was possible to set the region bounded by the minimum and maximum values observed for the parameter as the interval.

### 2.6. Uncertainty and Sensitivity Analyses

Uncertainty analysis (UA) and sensitivity analysis (SA) were conducted to assess and quantify the uncertainty spread from the unknown parameters to the model output, taking into account the effects of the interactions among those factors [35,36]. In this work, we treated UA as a graphical assessment of uncertainty propagation based on Monte Carlo (MC) simulation with parameter values sampled from previously defined ranges using a Latin hypercube design; we refer the reader to [37] for further information about this technique. We facilitated interpretation of the UA results by using the MC filtering approach described by Saltelli et al. [38]. In general, the MC filtering procedure can be outlined as follows:Split the MC simulated outputs into two groups, according to some property. In our specific case, we initially selected a single model output ($Y_{i}\left(t,\theta \right),t\in 0,1,\cdots ,k$) and used the estimated set of parameters representing the current configuration of the Colombian cancer healthcare system ($\hat{\theta }$), as determined following the methodology described in Section 2.5, in order to define the following membership function:
(3)$$\xi \left(\theta \right)=\left\{\begin{array}{cc} 1 & \;\mathrm{if}\;\sum_{j=1}^{k}Y_{i}\left(t,\theta \right)\geq \sum_{j=1}^{k}Y_{i}\left(t,\hat{\theta }\right)\\ 0 & \;\mathrm{otherwise}.\; \end{array}\right.$$The membership function allows us to split the set of parameter values (Θ) into two sets: A set of higher values $\theta^{1}:=\left\{\theta \in \Theta :\xi \left(\theta \right)=1\right\}$ and a set of lower values $\theta^{0}:=\left\{\theta \in \Theta :\xi \left(\theta \right)=0\right\}$. These sets, in turn, induce a marginal splitting of the values for each parameter ($\theta_{i}$) into two groups $\theta_{i}^{1}$ and $\theta_{i}^{0}$. This last marginal splitting can be used to analyze the influence of each parameter on the behavior of the model output according to the property we chose in the first step.Contrast the shape of the empirical cumulative distribution function (eCDF) for marginal higher values ($\theta_{i}^{1}$), marginal lower values ($\theta_{i}^{0}$), and marginal values ($\theta_{i}$), plotting them together in a single graph for each parameter individually. The eCDF of $\theta_{i}$ serves as a prior. The relevance of a parameter for the model output becomes more significant when there is a greater disparity between the behavior of the empirical cumulative distribution function (eCDF) for $\theta_{i}^{1}$ and $\theta_{i}^{0}$ with respect to $\theta_{i}$. Additionally, it is possible to identify parameter values with a high likelihood of causing the model output to exhibit any of the properties specified in the first step.

We also performed SA to quantify the relevance of each parameter with regard to the behavior of the model output. We chose a global approach for SA instead of a local one, as the former attempts to quantify the contributions of the model parameters in their entire distribution range, while the latter is only informative for a single set of values [38]. Furthermore, we chose the variance-based multivariate method proposed by Xiao et al. [39] to compute sensitivity indices, due to both the acceptance of such methods by the scientific community [38] and the multivariate approach of this technique.

### 2.7. Economic Tools

For the economic analysis of the model outputs, the initial step involves examining the patterns in cancer expenditures and mortality. The former is assessed by considering its relationship with the Gross Domestic Product of the country (CE) and population (per capita), whereas the latter is evaluated both in aggregate and per 100,000 inhabitants (MR, mortality rate). These assessments were performed for two timeframes: the 8-year period from 2022 to 2030, and the 10-year span from 2031 to 2040. The comparison between different scenarios ($s_{i}$) and a baseline (bl) was made using the growth rate (gr) of MR and CE, as shown in Equation (Equation 4) below, where $e_{s_{i}/bl}$ refers to the elasticity of scenario *i* with respect to the baseline.
(4)$$e_{s_{i}/bl}=\frac{grMR_{s_{i}}/grMR_{bl}}{grCE_{s_{i}}/grCE_{bl}}.$$

In the discussion, a financial analysis is presented by discounting the expenditure flows for the two horizons to present value. For scenarios entailing a permanent increase in spending, the annual ratio of financing (FR) is calculated relative to the baseline. To calculate the net present value, the expenditure flows are discounted using a social discount rate (δ) of 9%, as recommended by the National Planning Department [40] for the socioeconomic evaluation of projects in Colombia. The percentage difference between the net present values of each scenario ($NPV_{s_{i}}$) and the baseline ($NPV_{bl}$) is then estimated using Equation (Equation 5): (5)$$FR_{s_{i}}=\frac{NPV_{s_{i}}-NPV_{bl}}{NPV_{bl}}=\frac{\sum_{t=0}^{n}\frac{CE_{s_{i}}}{{\left(1+\delta \right)}^{t}}-\sum_{t=0}^{n}\frac{CE_{bl}}{{\left(1+\delta \right)}^{t}}}{\sum_{t=0}^{n}\frac{CE_{bl}}{{\left(1+\delta \right)}^{t}}}.$$

For scenarios where the changes are considered a long-term investment (i.e., lower expenditure with respect to the baseline), the financial analysis includes an estimate of the internal rate of return (IRR) for each of the time horizons, as returns are expected with respect to the baseline expenditure. To estimate the IRR, the difference in expenditure is calculated for each period ($\Delta CE_{t}$), and the flows of gains or cancer expenditure losses are identified ($\Delta CE_{t}$) using Equation (Equation 6): (6)$$\Delta CE_{t}=CE_{bl,t}-CE_{s_{i},t}.$$

Subsequently, the IRR is computed as the discount rate, which equates the present value of inflows ($PVI_{s_{i}}$) to the present value of outflows ($PVO_{s_{i}}$). In other words, it is the rate that renders the net present value ($NPV_{s_{i}}$) of the income and expenditure flows above the baseline equal to zero, as illustrated in Equation (Equation 7): (7)$$NPV_{s_{i}}=PVI_{s_{i}}-PVO_{s_{i}}=\sum_{t=0}^{n}\frac{\Delta CE_{s_{i},t}}{{\left(1+IRR_{s_{i}}\right)}^{t}}=0.$$

Utilizing these indicators enables assessment of the long-term impact of any strategy that seeks to modify the prevailing conditions in the treatment of cancer, with regard to the rate of change observed (compared to baseline).

## 3. Results

### 3.1. Schematized System

Figure 1 summarizes the natural history of the disease alongside the characteristics of the Colombian health system. We identified several processes that can be treated as subsystems, such as the population dynamics, the generation of new cancer cases, the evolution of the disease, the identification and treatment of those new cases by the health system, and the components in the health system acting as a manager of the limited public resources. We also identified four major outputs that help to assess the performance of the health system as a whole when dealing with cancer: incidence, survival rate, mortality rate, and spending.

We found ourselves unable to model some of the components of the summarized system in Figure 1 due to a lack of available information. For instance, we were aware of the active role of the mediation between the EPS and the IPS to give the patients a good treatment while trying to balance a system with limited resources (see payment model in Figure 1); nevertheless, to schematize such a process, developing appropriate equations and parameters for modeling would require in-depth knowledge about the functioning of the EPS and the IPS, which is mostly private information. Consequently, we directed our focus toward the components that we could feasibly model, thus ensuring the reliability of their inferred parameters. This applied to the process of generating new disease cases, the evolution of the disease for patients in relation to the four stages we identified, and the process of identification and treatment. In modeling these aspects, we took into account the relevance of age groups for this particular disease.

We started by looking for a model of the population dynamics of the country that ranges from some years in the past to the future of our current situation. Years in the past give us sufficient time to reach a stationary close state for the rest of the components in the model, whose variation only depends on the changes in the population structure (and, maybe, some new health-related policies). In this way, we were able to bypass estimation of all of the initial conditions for our model. We finally chose the time-series data freely provided by DANE for the past and future population of Colombia, as well as their decomposition into age groups, from the year 2005 up to the year 2070. We refer the reader to [21] for further information about these time-series. As DANE estimates population annually instead of monthly, we decided to keep the total population of the model fixed annually.

### 3.2. Mathematical Model

From the schematized system in Figure 1, focusing on the information available and the natural history of the disease, we managed to propose the mathematical model depicted in Figure 2. This model retains the components we were able to parameterize, calibrate, and validate after setting up Assumptions 1–10:

**Assumption** **1.**
*The number of people with the disease is irrelevant to the population dynamics (i.e., there is no feedback between the disease-related components and the population dynamics model).*


**Assumption** **2.**
*Cancer can be modeled as a single disease, despite it being a cluster of diseases with remarkable differences among them (nevertheless, it is possible to set up the same mathematical structure proposed here to model each particular cancer as an isolated disease or a set of diseases).*


**Assumption** **3.**
*People with the disease only die after reaching stage IV (i.e., the stages in the model are at the same time related to the classification of the disease and its severity).*


**Assumption** **4.**
*The evolution of the disease is independent of age and gender for both patients with and without treatment.*


**Assumption** **5.**
*The cost of the procedures and medicines involved in the treatment is independent of the age and the gender of the patient.*


**Assumption** **6.**
*There are different qualities of treatment (i.e., the patients are treated unequally).*


**Assumption** **7.**
*The quality of the treatment for a patient does not change over time, or because of the number of patients being treated.*


**Assumption** **8.**
*The health system has infinite capacity for cancer care.*


**Assumption** **9.**
*The quality of the treatment is positively correlated with the cost of the treatment.*


**Assumption** **10.**
*All undiagnosed people that enter stage IV become diagnosed at the next time step.*


**Figure 2 ijerph-20-06740-f002:**
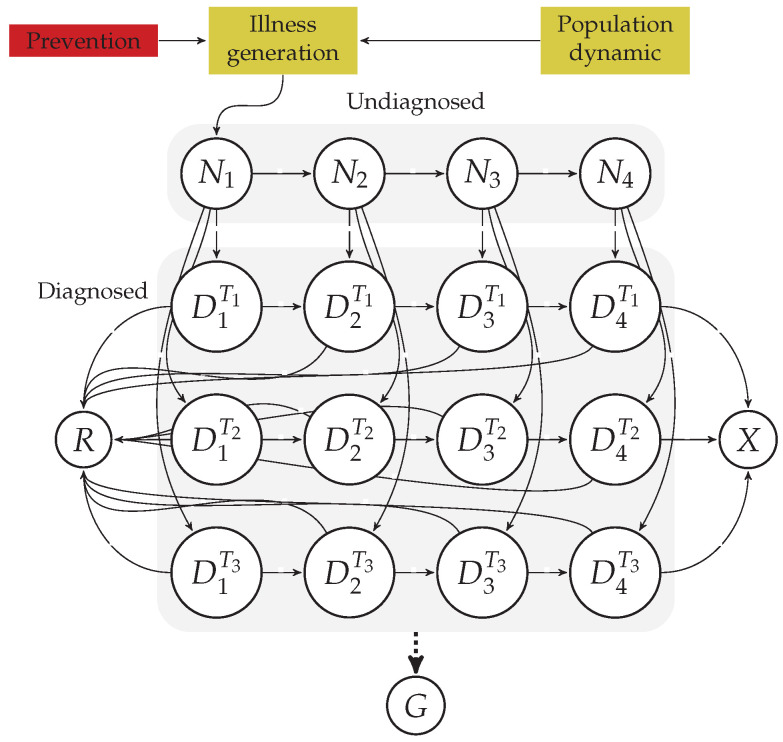
Diagram of the mathematical model inferred from schema in Figure 1 with three different treatment quality levels ($T_{1}$–$T_{3}$). The colored boxes at the top of the figure represent model components that we do not model through states. Most of the states can be grouped, as they have or have not received attention from the cancer healthcare system (diagnosed and undiagnosed ones, respectively). The natural model outputs are the patients who die at every time step, X(t), the patients who recover from the disease at every time step, R(t), and the total spending on cancer healthcare at every time step, G(t). The dotted line for *G* indicates that there is no inflow of individuals into *G*; instead, *G* receives an inflow in currency, induced by the number of individuals in the diagnosed states.

Considering the natural history of the disease in the Colombian context, we found all of our assumptions—except for Assumptions 4 and 8—to be reasonably valid. However, we lack the necessary information to address these assumptions. Overcoming Assumption 4 would require reliable data on disease progression, specific procedures performed, medications administered, and detailed costs categorized by age group and gender. It is important to note that, despite Assumption 5 being justified by the observed minimal influence of age and gender on procedure costs [41], Assumptions 4 and 5 collectively result in overall treatment costs being independent of the age and gender of the patients. On the other hand, overcoming Assumptions 8 and 7 would necessitate knowledge about the capacity of the healthcare system and its behavior as it reaches saturation. However, these Assumptions are not relevant for the identification and modeling of the current configuration of the Colombian cancer healthcare system. Nonetheless, they may gain relevance if the parameters are modified to simulate hypothetical scenarios wherein the real system could become overloaded. Finally, we chose months as our simulation time unit, as this gives time enough for a patient to be diagnosed or treated, and also gives sufficient time for the disease to progress (i.e., it is unnecessary to consider time delays in the mathematical structure, as they will be naturally included).

#### 3.2.1. The State of Undiagnosed Patients

The proposed model starts every time step by considering the population structure of the country. Then, according the age-related incidence rates of cancer from GLOBOCAN, some individuals develop the disease. Each individual with the disease starts their cycle in the compartment for undiagnosed patients at stage I ($N_{1}$). From now on, we represent the stage of the disease for the compartments (where it is relevant) with a subindex. In the next time-step, the undiagnosed patients may be detected (i.e., they become diagnosed patients) or not, according to a probability of detection depending on the stage of the patient. The patients that remain undiagnosed can, in turn, remain in the same stage or progress to the next one, according to a stage-related probability. To meet Assumption 10, an undiagnosed patient in stage 4 ($N_{4}$) can only become diagnosed in the next time step. Denoting detection probabilities using the Greek letter δ and progress probabilities using the Greek letter α, we use the following representations for all undiagnosed states: (8)$$\begin{array}{cc} N_{1}\left(t+1\right) & =\left(1-\delta_{0}\right)I\left(t\right)+\left(1-\delta_{1}\right)\left(1-\alpha_{1}\right)N_{1}\left(t\right),\\ \; & \vdots \\ N_{4}\left(t+1\right) & =\left(1-\delta_{3}\right)\alpha_{3}N_{3}\left(t\right). \end{array}$$

Note that the input of state $N_{1}$ (I(t)) in the system of Equation (Equation 8) is modified by the parameter $\delta_{0}$. The purpose of this parameter is to model very early detection (i.e., patients that are detected almost immediately after developing the disease). As the reader can see in the next section, the remaining input of $N_{1}$ is focused on the diagnosed states; see the system of Equation (Equation 9).

#### 3.2.2. The State of Diagnosed Patients

According to the data available from CAC regarding the degree of fulfillment of government-recommended good practices for cancer treatment, there exists heterogeneity in the quality of attention each patient receives, seemingly due to variations in equipment, capacity, and the availability of qualified personnel in different regions [18]. We incorporated this heterogeneity into the overall quality of attention by defining three treatment levels, categorized based on their effectiveness: $T_{1}$ for the lowest quality level, $T_{2}$ for an intermediate level, and $T_{3}$ for the highest quality level. Therefore, when a patient is diagnosed, they begin receiving treatment at one of these defined quality levels, which remains consistent throughout their journey within the cancer healthcare system. It is important to note that the quality of treatment in Colombia is more dependent on geographical factors, rather than being a random variable. However, due to the current state of development of the model, we were unable to incorporate such a spatial dependence. Similar to the case of stages, we denote the type of treatment a patient receives by adding a superscript to the state; for example, $D_{i}^{T_{j}}\left(t\right)$ represents the number of diagnosed patients in stage *i* receiving treatment *j* at time step *t*.

When a patient is diagnosed, they retain the same stage they have at the current time-step. Additionally, due to Assumption 7, the treatment of the patient does not change over time and, so, their only options are to recover or progress to the next stage (up to stage IV). Once a patient reaches stage IV, they may also die because of the disease. Setting $\lambda_{j}^{i}$ as the probability of receiving treatment *i* after being detected in stage *j*, $\gamma_{i}^{i}$ as the recovery probability, and $\alpha_{i}^{i}$ as the progression probability (with $\alpha_{4}$ being the probability of death), we obtain the following representation for all diagnosed states associated with treatment quality *i*: (9)$$\begin{array}{cc} D_{1}^{T_{i}}\left(t+1\right) & =\delta_{0}\lambda_{1}^{i}I\left(t\right)+\delta_{1}\lambda_{1}^{i}N_{1}\left(t\right)+\left(1-\gamma_{1}^{i}\right)\left(1-\alpha_{1}^{i}\right)D_{1}^{T_{i}}\left(t\right)\\ \; & \vdots \\ D_{4}^{T_{i}}\left(t+1\right) & =\lambda_{4}^{i}N_{4}\left(t\right)+\left(1-\gamma_{3}^{i}\right)\alpha_{3}^{i}D_{3}^{T_{i}}\left(t\right)+\left(1-\gamma_{4}^{i}\right)\left(1-\alpha_{4}^{i}\right)D_{4}^{T_{i}}\left(t\right). \end{array}$$

#### 3.2.3. The Model for Age Groups

As we incorporate the influence of age groups into the dynamics of the disease, we arrive at a more precise model representation, as shown in Figure 3. In this final model, we divided each stage into age groups with a one-year difference, ranging from 0 years old to 100+ years old, based on the available data from DANE [21]. However, the age-related incidence data from GLOBOCAN utilizes age groups with a span of 10 years (e.g., 0 to 9, 10 to 19, and so on). Therefore, we opted to adopt the cancer incidence values from GLOBOCAN for each age group within that range (i.e., the same incidence from 0 to 9 and so on). Alternatively, we could have used interpolation techniques to assign incidences to our age groups; however, this would have required making additional assumptions about the incidence behavior.

It should be noted that every panel of the age groups shown in Figure 3 almost contains the whole model shown in Figure 2. This is because modeling age groups for this case is the same as proposing the modeling of meta-populations sharing the same underlying model [17]. As states for individuals that become recovered (R(t)), individuals who died (X(t)), and spending (G(t)) can be described as a linear combination of diagnosed states, we can conclude that our whole model is linear at its current state of development. However, defining a single transition matrix to simulate the model as a Markov process is a time-demanding and unnecessary process that may hinder the suitability of the model for the simulation of public policies. Instead, we decided to define a clear order for the processes involved in the evolution of the system and defined simpler intermediate transition matrices, as described in the following.

First, it is useful to note that a simple linear equation, such as the first one in the system of Equation (Equation 8), can be simplified even further after performing the consecutive change of variables $N_{1}^{*}(t)=\left(1-\delta_{1}\right)N_{1}(t)$ and $N_{1}^{+}(t)=\left(1-\alpha_{1}\right)N_{1}^{*}(t)$, as we obtain that $N_{1}(t+1)=\left(1-\delta_{0}\right)I(t)+N_{1}^{+}(t)$. Such substitutions can be perceived as meaningless, but they are not. As stated in the Methodology section, the modeling approach we followed requires of the definition of an order or hierarchy of flows outside any state. Hence, we chose the natural hierarchy of recovery > detection > stage progress/death. In this order of ideas, note that $N_{1}^{*}(t)$ represent the number of people undiagnosed in time step *t*, while $N_{1}^{+}(t)$ represent the number of people that neither progress to the next stage nor become diagnosed ones; that is, the number of people in $N_{1}(t)$ remaining in that very state. Then, from an algorithmic point of view, the following assignments are correct and lead to the same result as the original equation: $$\begin{array}{cc} N_{1}(t) & ⟵\left(1-\delta_{1}\right)N_{1}(t)\\ N_{1}(t) & ⟵\left(1-\alpha_{1}\right)N_{1}(t)\\ N_{1}\left(t+1\right) & ⟵\left(1-\delta_{0}\right)I(t)+N_{1}(t). \end{array}$$

Following the same idea, in order to simulate the whole model, it suffices to create two matrices: one to contain all the states related to diagnosed patients and another to create the same for the undiagnosed ones Then, it only remains to build up appropriate matrices to model the three major actions stated above. As a result, we reached an intuitive and simple model implementation, which the reader can consult in the online GitHub repository https://github.com/drojasd/MDPICancerPolicies (accessed on 27 February 2023). We selected the number of individuals that recovered at time *t* (R(t)), the number of individuals that die because of the disease at time *t* (X(t)), and the total spending on cancer diagnosis and treatment at time *t* (G(t)) to be the main model outputs. Furthermore, we also proposed the number of undiagnosed and diagnosed patients as model outputs, both with respect to stage and age group.

### 3.3. Parameter Identification

To calibrate the model, we considered it necessary to estimate some parameters that could not be obtained or inferred directly from the literature. In the following sections related to estimation, we detail the strategies that we followed to reliably identify these parameters. On the other hand, the last subsection is related to the process of identification for those parameters that did not require estimation (i.e., identification without optimizing a mathematical model).

#### 3.3.1. Estimation of Detection Probabilities and Stage Progression for Undiagnosed Compartments

To model the states of undiagnosed patients, we needed to estimate parameters for inflows and outflows at every state. As indicated in Section 3.2.1, except for the early diagnosed patients, all cancer patients start their path in the state $N_{1}$. Undiagnosed patients can be detected (diagnosed) or, if not, they could progress to the next stage as their health status worsens. Hence, we need to identify seven parameters for this model: four probabilities of detection (δ) and three probabilities of disease progression (α). Figure 4 displays the structure for undiagnosed patients as an independent model.

Data from CAC for the year 2021 [18] revealed that 8.74% of cancer cases are detected at their very start, 18.76% are detected at stage I, 24.69% at stage II, 23.65% at stage III, and 24.14% at stage IV. However, such values are not necessarily equivalent to the required values of $\delta_{0}$–$\delta_{3}$. From Figure 4, note that the volume of patients detected at a certain time step *t*, D(t), is given by: $$D\left(t\right)=I\left(t\right)\delta_{0}+\sum_{j=i}^{3}N_{j}\left(t\right)\delta_{j}+N_{4}\left(t\right).$$

Then, we can state that $\frac{I\left(t\right)\delta_{0}}{D(t)}=0.0874$, $\frac{N_{1}\left(t\right)\delta_{1}}{D(t)}=0.1876$, and so on. Furthermore, assuming that the cancer healthcare system is close enough to a stationary state (i.e., all of the external factors that influence the incidence, diagnosis, and treatment of cancer change slowly with the time), we can set the net inflow to the undiagnosed state equal to its outflow I(t)=D(t)=D. In that case, it would be immediate that $\delta_{0}=0.0874$, but the other six parameters remain unknown and the resultant equations generate a system of nonlinear equations. To solve such a problem, we implemented the model in Figure 4 using the GSUA_CSB toolbox, as described in Section 2.5. This model served as an auxiliary model for parameter estimation.

To estimate the parameters in Table 1, we first identified a time horizon sufficient to assure that the auxiliary model reached the stationary state through MC simulation with the estimation ranges of the parameters. We found that t>90 holds for the condition above. Then, we chose an arbitrary positive constant for the input function I(t)=1e5, set the model outputs in the form $\left[I\left(t\right)\delta_{0},\cdots ,N_{i}\left(t\right)/D\left(t\right),\cdots ,N_{4}\left(t\right)/D\left(t\right)\right]$, and fit those outputs to horizontal lines fixed at the values provided by CAC. We repeated the fitting procedure 1000 times, starting the optimization algorithm from different random points inside the estimation ranges. Then, as suggested in the methodology, we selected the 10% of the estimates with the best fit to the data and set the estimated intervals for the parameters using the minimum and maximum values of those filtered estimates. We selected the estimate with the best fit to the data as a nominal estimate (see Table 1). A more detailed validation of the estimations is provided in Appendix B.

#### 3.3.2. Estimation of Stage Transitions and Recovery for Detected Compartments

Similarly to the case of parameters involved in the transitions and outflows of the undiagnosed compartments, we found ourselves unable to use the information about cancer survival and mortality to infer the values of the parameters in the model in a straightforward way. As noted in Section 3.2.2, all of the parameters related to recover from the disease (γ) and disease progression (α) are related to the current stage of the patient and the treatment quality level.

The first challenge we met was the fact that each type of cancer has its own five-year survival percentage, but we overcame this by computing a weighted mean of the available marginal survivals (see Table 2). We assumed that the variation in the five-year survival percentage for each of the three treatment quality levels proposed in Section 3.2.2 was the same as that evidenced for the largest cluster of countries found in the GLOBOCAN data (i.e., Australia, Canada, Denmark, Ireland, Norway, and the UK). In this way, we were able to determine the three different five-year survival percentages needed for our model.

The next step was to propose the auxiliary model, presented in Figure 5, which is used to estimate recovery and disease progression probabilities. The structure of this auxiliary model was designed to allow us to identify the percentage of patients that do recover or die from the disease after some time, with regard to the extent they are detected at one of the different stages. Both the four recovery parameters ($\gamma_{1}$–$\gamma_{4}$) and the four progression parameters ($\alpha_{1}$–$\alpha_{4}$) appear several times in the structure of the auxiliary model (see Figure 5), thus increasing their identifiability.

Finally, it was necessary to assume that, after a certain duration of treatment, the majority of patients either recover or pass away. We selected this duration as one year, meaning that most patients in our model recover or pass away within one year of treatment. While this assumption may be considered strong, it is grounded in the fundamental idea that a significant portion of the healthcare expenses associated with cancer treatment occurs during the initial year. The estimation process for each level of treatment quality required a separate analysis. For each estimate, we assigned an arbitrary (but fixed) positive number of patients as the initial condition for the states $D_{1}^{1},D_{2}^{2},D_{3}^{3},$ and $D_{4}^{4}$, representing cohorts of patients diagnosed at each stage. Subsequently, we allowed the model to run for 12 time steps (equivalent to one year) and fitted its outputs ($R^{1}$–$R^{4}$) to linear trends representing the expected number of survivors based on the five-year survival percentage for the respective treatment quality level. To account for the possibility of some individuals continuing their treatment beyond the first year, we fitted the model to linear trends that extended from 99% to 100% of the expected number over a one-year period. This approach allowed for a total treatment duration of two years, accommodating patients who may require ongoing care after the initial year.

In Table 3, we summarize the estimated parameters related to disease recovery and progression for each treatment quality level. We conducted almost exactly the same estimation procedure described in Section 3.3.1 to obtain Table 1. The only difference is that we performed 500 estimations instead of 1000 estimations per each identification procedure. Note that we set nominal undiagnosed progressions of Table 1 as superior limits for the estimation intervals of diagnosed progressions in Table 3. We support this decision, as one can expect the health status of patients without treatment to become worse faster than that of those under treatment. The reader can refer to Appendix C for details regarding validation of the estimated parameters.

It is worth noting that the recovery probabilities estimated for stages I and III exhibited an unexpected behavior: they increased as the treatment quality worsened, as indicated by the values for $\gamma [s_{1}\cdot ]$ and $\gamma [s_{3}\cdot ]$ in Table 3. Another intriguing observation pertains to the progression probabilities for all stages except death, as they were constrained to their upper bounds during estimation, as evident from the values for α in Table 3. In both of these cases, it is important to consider that the estimation algorithm was allowed to produce such results as no constraints were imposed during optimization other than ensuring that the values remained within the designated estimation ranges in Table 3. Consequently, the parameter values obtained represent the best fit for explaining the behavior of the system regarding patient mortality and five-year survival. For example, the similarity between progression probabilities before and after diagnosis, combined with low recovery probabilities at early stages, suggests the presence of delays in initiating treatment following diagnosis, as elaborated upon later in the Discussion.

#### 3.3.3. Estimation of Probabilities for Treatment Quality and Treatment Cost

To identify the probability of a patient receiving a certain treatment quality level, we used data available from CAC related to the characterization of the patients according to the degree of fulfillment of government-recommended good practices [18]. From such data, we determined that approximately 19.6% of the patients received deficient attention, 20.9% received intermediate attention, and 59.6% received good attention. Furthermore, we found information suggesting the quality of the treatment to be more linked to geographical conditions than the stage of disease of the patient [18]. Thus, we decided probabilities related to treatment quality levels (λ) should be independent of the stage of the disease. Additionally, as all the patients received one and only one level of quality in the treatment, there were only two degrees of freedom and it is possible to remove one of the parameters. Finally, to avoid issues in simulation, it is necessary to define a hierarchy for treatment quality assignment, such as the one detailed in Example 1. Furthermore, according to information from CAC, the values for the parameters are as follows: $\lambda [s_{*}T_{1}]=0.196$ and $\lambda [s_{*}T_{2}]=0.260$.

**Example** **1.**
*Input flow to diagnosed stage I states after setting up the hierarchy $\lambda^{T_{1}}>\lambda^{T_{2}}>\lambda^{T_{3}}$ for treatment quality assignment.*

$$\begin{array}{cc} D_{1}^{T_{1}}\left(t+1\right) & =\delta_{0}\lambda_{1}^{1}I\left(t\right)+\cdots \\ D_{1}^{T_{2}}\left(t+1\right) & =\delta_{0}\left(1-\lambda_{1}^{1}\right)\lambda_{1}^{2}I\left(t\right)+\cdots \\ D_{1}^{T_{3}}\left(t+1\right) & =\delta_{0}\left(1-\lambda_{1}^{1}-\left(1-\lambda_{1}^{1}\right)\lambda_{1}^{2}\right)I\left(t\right)+\cdots . \end{array}$$



On the other hand, we used the work of Gamboa et al. [30], who estimated the cost of breast cancer treatment for the year 2012 in the Colombian case, given the stage at which the patient is diagnosed, in order to obtain the expected monthly costs of cancer treatment at any date in the chosen simulation period (2005–2070). We took into account the effects of inflation and the expected growth of the GDP for the country to make available some interesting economical outputs for the model. Table 4 provides the monthly costs inferred from the work of Gamboa et al. [30]. Furthermore, considering Assumption 9, we assigned a lower bound for the estimation of costs in [30] to lower-quality treatment ($T_{1}$), and so on.

In addition to the parameters linked to the cost of treatment, we included a parameter for the cost of diagnosing a patient, $C_{\delta }$, and took its value to be the mean cost of cancer diagnostic-related procedures reported in the SISPRO database. In this way, we covered all cancer treatment costs. However, there is an issue that arises when using data from breast cancer alone to identify the average cost of treatment for all cancer types as a single entity, due to the high heterogeneity in cancer treatment costs according to the type of disease. To overcome this issue, we used the data available in the SISPRO database for Colombian prioritized cancer types (see Figure A1 in Appendix A) related to both the participation of each cancer type in the total spending and the number of patients per cancer type, in order to estimate a correction factor ϵ. This correction factor determines the error obtained when assigning the average cost for breast cancer to all cancers.

The correction factor ϵ can be computed as follows. Let $C_{b}$ be the average spending per breast cancer patient, $C_{T}$ be the average spending per patient for any of other type of cancer, *C* be the average spending per patient for any cancer, and *B* be the proportion of breast cancer patients. It is immediately obvious that $C=C_{b}B+C_{T}\left(1-B\right)$. Then, there exists a constant ϵ such that $C_{b}B+C_{T}\left(1-B\right)=\epsilon C_{b}$. It follows that
(10)$$\epsilon =\frac{C_{b}B+C_{T}\left(1-B\right)}{C_{b}}=\frac{C}{C_{b}}.$$

Replacing the variables in this last equation with the information from SISPRO (summarized in Figure A1), we obtained ϵ≈0.642. Note that applying ϵ to any of the costs in Table 4 is equivalent to computing $\epsilon C_{b}=\frac{C}{C_{b}}C_{b}=C$, which successfully corrects for the bias.

### 3.4. Model Validation

After estimating all of the parameters above, we were ready to implement and simulate the model in Figure 3. Note that this model was never fitted to real data. Even when the components of the model were tuned using information from several sources, the information about cancer mortality rate and total spending in cancer were not used until this point because, as exposed in Section 2.5, we selected them as control output variables to assess the suitability of the estimated parameters.

We obtained information regarding the total spending on cancer for the years 2015–2018 from the work of Restrepo-Zea et al. [43], who used information from two government-related entities—ADRES (Resources of the General System of Social Security in Health Administrator) and SISMED (System of Information of Prices in medicines), by their respective Spanish acronyms—to estimate costs, as well as data from CAC to obtain the prevalence of the disease for the same period. Furthermore, Restrepo-Zea et al. [43] reported a strong linear correlation between the prevalence of and total spending on cancer. Thus, we decided to take advantage of this correlation to prolong the spending time-series up to the year 2021, as CAC provides prevalence data for the remaining dates.

As shown in Figure 6, we assessed the reliability of the model by comparing its outputs for mortality and spending against the available real-world data. The strong resemblance between the model outputs and the real data suggests that the methodology we employed to estimate and infer the model parameters was suitable for this case. In other words, we successfully identified a model that accurately represents the cycle of cancer treatment within the Colombian health system. Notably, the high degree of similarity in the spending graph is a significant result, as it supports our assumption regarding the maximum treatment time (which will be discussed later). Furthermore, the sharp increase in mortality data observed after 2019, as well as its notable deviation from the baseline model mortality, will also be addressed in subsequent discussions.

### 3.5. Modeling of Disease

After we successfully identified a model for the treatment of cancer patients in the Colombian health system, we moved forward by integrating mathematical components into the model structure aiming to simulate the implementation of relevant public policies. We also explored the effects of the parameters on the behavior of the model through sensitivity analysis and MC filtering, in order to identify key parameters. Such key parameters provide relevant information about the components of the health system that can be focused on to improve outcomes.

#### 3.5.1. Modeling of Prevention and Policy Implementation

First, we wanted to simulate the speed of the transition when implementing or changing public policies in the health model. Thus, we included a switch inside the model that triggers a change in certain parameter values when the simulation time surpasses a threshold value. Let τ>1 be the number of time steps elapsed after the simulation time reaches the threshold. We define the transition function β(τ) as
$$\beta \left(\tau \right)=\beta_{1}^{1/(\beta_{2}\tau )},$$
where $\beta_{1}\in \left[0,1\right]$ and $\beta_{2}>0$ are dimensionless transition parameters controlling the transition. Note that, for $\beta_{1}$ and $\beta_{2}$ close enough to 0, it is possible to obtain β(1) as close to 0 as desired. On the other hand, we have that $\lim_{\tau ⟶\infty }\beta \left(\tau \right)=1$. Further, note that the value for $\beta_{1}$ controls how close β(τ) is to 1 for the lowest values of τ, while $\beta_{2}$ controls the speed of convergence. In terms of public policies, $\beta_{1}$ models the abruptness of the change while $\beta_{2}$ models the time horizon to complete the change. Example 2 illustrates how we used β(τ) to model the change in the parameters of the system. We performed almost the same thing, but defined time-dependent transition matrices instead of functions.

**Example** **2.**
*After setting up a break-point time ($t^{*}$) where the change in public policies starts, all the parameters in the model become time-dependent. For instance, suppose we have the time-dependent parameter δ(t) whose value before the change in policies is δ. The objective of the new policies is to shift the value of δ(t) from δ to $\delta^{*}$, according to some strategy modeled by β(τ). Then, δ(t) must be described as*

$$\delta \left(t\right)=\left\{\begin{array}{cc} \delta & \;\mathrm{if}\;t\leq t^{*}\\ \delta +\left(\delta^{*}-\delta \right)\beta \left(t-t^{*}\right) & \;\mathrm{if}\;t>t^{*}. \end{array}\right.$$



As suggested by Figure 1, Figure 2 and Figure 3, prevention plays a major role in the treatment of cancer. Moreover, the WHO [44] estimated that between 30–50% of all cancer cases are preventable. To model prevention, we used the fact that the input function of new cancer cases has the form $I\left(t\right)=\sum_{i=1}^{w}g_{i}\left(t\right)\phi_{i}$, where *w* is the number of age groups, $g_{i}\left(t\right)$ is the number of people in the ith age group, and $\phi_{i}$ is the proportion of individuals getting cancer in the ith age group at the current time step. Then, we propose $I\left(t\right)=\sum_{i=1}^{w}g_{i}\left(t\right)\phi_{i}\left(t\right)$, where
$$\phi_{i}\left(t\right)=\left\{\begin{array}{cc} \phi_{i} & \;\mathrm{if}\;t\leq t^{*}\\ \phi_{i}^{1+\eta \beta (t-t^{*})} & \;\mathrm{if}\;t>t^{*}, \end{array}\right.$$
in which η≥0 is the (dimensionless) prevention parameter. We chose this structure to model prevention as cancer is an age-related disease that becomes more likely as the age of the individual increases. Thus, it should be more easy to prevent cancer development in younger individuals than in older ones, which is exactly the behavior modeled by $\phi_{i}\left(t\right)$.

Taking into account the three parameters that were added to model preventive efforts and the implementation of public policies, we obtained a total of 55 parameters. We decided to explore the effects of these parameters except the ones related to costs (12 of them) on the behavior of the model for cancer spending and deaths. To achieve such a goal, we first proposed an appropriate output to be used as target for sensitivity and uncertainty analyses. Hence, we took the output of total spending in cancer at time *t* (G(t)) and the output for the number of deaths at time *t* (X(t)), both divided by the total size of the population at the same time (P(t)), and multiplied them to obtain the target output ($T\left(t\right)=\frac{G(t)}{P(t)}\frac{X(t)}{P(t)}$). Dimensional analysis shows the units of *T* to be
$$\left[T\right]=\frac{\left[G\right]}{\left[P\right]}\frac{\left[X\right]}{\left[P\right]}=\frac{\left[G][X\right]}{\left[P\right]}\frac{1}{\left[P\right]},$$
where [G] are the units of the cancer spending (Colombian currency, COP), [P] the units of total individuals, and [X] the units of individuals who died. Clearly, as [G]/[P] is the cancer cost per capita (i.e., the cancer cost assumed by each citizen), $\frac{\left[G\right]}{\left[P\right]}\left[X\right]$ would be the cancer cost assumed by people who died. However, people who died cannot assume any cost, causing such cost to be redistributed over the remaining population. Assuming that P(t)/(P(t)−X(t))≈1, it follows that T(t) is approximately equal to the extra cost assumed by each individual due to cancer-caused deaths.

We believe this target to be appropriate for two main reasons. The first is that, regarding some base scenario, lower values for *T* should be linked to a better efficiency in the system while higher values should be linked to scenarios with lower efficiency than the base one. The second is the magnitudes of both of the outputs multiplied in *T* to be inversely the same; that is, G(t)/P(t)≈P(t)/X(t). This behavior causes a reduction in cancer spending to be as relevant as the reduction in the mortality rate. The target output has two extreme ideal scenarios: when no one dies from cancer and when there is no money allocated to cancer healthcare. To avoid the first extreme scenario, we shrunk the ranges for most of the parameters from their estimation ranges to 50% of their nominal value, as the reader can see in Table 5. Furthermore, to avoid the second extreme scenario (as well as spurious interactions among transitions, disease recovery, and costs), we decided to keep the parameters associated with treatment costs fixed at their nominal values (see Table 4).

Table 5 summarizes the SA results, performed as stated in Section 2.6. We chose the range for the prevention parameter ($\left[\underline{\eta },\bar{\eta }\right]$) in such a way that the total area under the curve of ϕ(t) was reduced by 30% (the lower bound for the % of preventable cancer, according to the WHO) at the maximum effort of prevention (i.e., $0.7\sum_{i=1}^{w}\phi_{i}=\sum_{i=1}^{w}\phi_{i}^{1+\bar{\eta }}$). The minimum effort of prevention corresponded to an scenario without effort of prevention ($\underline{\eta }=0$). As expected, the results in Table 5 demonstrated that the most relevant factor for the behavior of the model was η, followed by some parameters related to the quality of treatment.

From the SA results, we selected those parameters whose total-order sensitivity index ($SST_{i}$) explained more than 1% of the output variance and proceeded to perform the UA analysis (see Figure 7) and the MC filtering for the UA (see Figure 8). As expected, the parameters with the higher $\%SST_{i}$ were the parameters with higher differences for the eCDF of their lower ($\theta_{i}^{0}$) and higher ($\theta_{i}^{1}$) values in the MC filtering. The UA in Figure 7 exposed the base scenario—namely, the scenario with the identified parameters for Colombia—to be more closer to the extreme inefficient scenarios (higher curves) than to the extreme efficient ones (lower curves), suggesting that there is room for improvement of the current cancer healthcare system. On the other hand, the MC filtering results in Figure 8 suggest that achieving efficient scenarios is more challenging than inefficient ones when focusing on one parameter at a time. This is evident from the eCDF plots, where the eCDFs for lower parameter values closely resemble the prior distribution. In other words, lower scenarios appear to be independent of the specific values of individual parameters, instead emerging as a result of interactions among multiple parameters.

#### 3.5.2. Simulation of Scenarios

The findings reported in the previous section motivated us to test several scenarios resulting from modifications in the values of certain parameters that are closely associated with strategies in the field of public policy. Our objective was to identify sustainable strategies that can yield improved outcomes in cancer healthcare; however, it is important to note that the current state of development of the proposed model does not allow for the simulation of public policies. This limitation arises from the recognition that public policies are complex and exert an impact on the healthcare system by modifying the functioning of specific components, reallocating resources within the system, or directly providing funding to enhance certain processes. Nonetheless, all the parameters in the model have clear meanings that can help to guide the strategies required to achieve their modification. The scenarios we simulated were based on the following straightforward hypotheses that have the potential to shape strategies and policies in the field of cancer healthcare:

**Hypothesis** **1.**
*An increase in early detection will result in a decrease of the total deaths and costs.*


**Hypothesis** **2.**
*Providing a better treatment to patients detected at early stages will result in a decrease in the total deaths and costs.*


**Hypothesis** **3.**
*Improving the effectiveness and expediency of better treatments will make their cost more affordable.*


We set $\beta_{1}=0.2$ and $\beta_{2}=1$ to simulate a rapid transition from old to new parameter values associated with a new policy or strategy. Initially, we proposed several distinct scenarios based on the previous hypotheses to test them individually and in combination. However, we discovered that the most significant findings could be summarized into three scenarios: starting with the first hypothesis alone, then incorporating the second one, and finally including all of them simultaneously. These scenarios are presented in the following.

Scenario 1. We directly tested Hypothesis 1 in this scenario by increasing the early detection rate $\delta_{0}⟶1$. However, we noticed this hypothesis to be false. Even when we assumed the health system capacity to be limitless (Assumption 8), detecting a patient at an early stage exerts pressure over the costs to avoid the patient from dying, causing the spending to be unaffordable. Without Assumption 8, this scenario will cause a collapse of the health system as early treatments are not good enough.Scenario 2. Considering what we found from Scenario 1, we decided to test a combination of Hypotheses 1 and 2 in Scenario 2. In particular, we made $\delta_{0}⟶1$, $\lambda [s_{1}T_{1}]⟶0.1$, and $\lambda [s_{1}T_{2}]⟶0.2$. This combination of parameters causes the system to assign better treatment to its patients in addition to high early detection. However, we noticed Hypothesis 2 to be false also, as it caused the costs to rise even more than for Scenario 1, even though it further reduced the number of deaths.Scenario 3. Considering what we found from Scenario 2, we decided to test a combination of Hypotheses 1, 2, and 3 in Scenario 3. In addition to modification of parameters made in Scenario 2, we set $\gamma [s_{1}T_{3}]⟶0.6$ and $\alpha [s_{1}T_{3}]⟶0.3$. For this scenario, we finally achieved a long-term reduction of costs and the lowest mortality among the tested scenarios. This is an interesting result (which is further discussed later), as it suggests that Hypothesis 3 is more pronounced when the conditions of Hypotheses 1 and 2 are also met, highlighting the potential for their combined effects.

Figure 9 displays the results for the three different scenarios, compared to the baseline induced by current public policies. While all strategies led to decreases in mortality, it is important to note that most of them also resulted in increased costs, which may be unsustainable for many countries. Even the base scenario demonstrated an unsustainable growth in costs. These findings are in alignment with those presented in Section 3.5.1, and it became evident that the only long-term efficient scenario (Scenario 3) requires a combination of simultaneous changes in several parameters to lower both mortality and costs. It is crucial to recognize that the results in this section provide valuable insight into different strategies for public policy, based on the tested hypotheses. However, it is essential to acknowledge that the feasibility of achieving such radical changes in parameter values for the baseline scenario may pose significant challenges. These scenarios primarily serve to offer insights into potential directions for public policies, and should be interpreted in light of the practical feasibility of implementing such drastic parameter modifications.

#### 3.5.3. Economic Analysis

Despite the fact that the scenarios we simulated may not be realistic, it is still noteworthy that even strategies that can lead to highly desirable efficiency in the system may require an initial investment period with a higher demand for resources. For instance, higher early detection and universal high-quality treatment are goals pursued by almost every cancer healthcare program. However, solely prioritizing an increase in early detection may lead to an influx of patients that overwhelms the system, potentially causing saturation. Our model, based on Assumption 8, represents this overflow as an unsustainable escalation in costs, as demonstrated in Figure A4 in the Appendix D. Therefore, achieving the goals of higher early detection and universal high quality would also require investments aimed at improving the available treatments and expanding the capacity of the current system. In light of these considerations, we decided to conduct a detailed analysis, from an economic perspective, of the results obtained from the simulation of Scenarios 1–3.

To evaluate the impact of each of the scenarios, the cost of cancer (% GDP) was used as an indicator of financial outcomes, while mortality was used as an indicator of health outcomes (per 100,000 population). The results are summarized in Table 6. The baseline scenario shows that, over a 20-year horizon, the cost of cancer will increase from 0.56% in 2022 to 0.87% in 2040 (equivalent to an annual growth of 3.04%), while the mortality rate will increase from 68.4 to 92.4 (an annual growth of 1.95%).

A progressive increase in early detection (Scenario 1) succeeds in reducing the mortality ratio to 62 in 2040, a 9.3% decrease from 68.4 in 2022. However, implementing this strategy without changing current treatment conditions will increase the cancer cost to 1.01% of GDP in 2040 (i.e., 0.14% above the baseline scenario). This implies that a 1% increase in cancer cost reduces the mortality rate by 2.15% over the baseline scenario, holding all else constant.

In Scenario 2, where the type of treatment is also improved, the mortality rate is further reduced to 56.7 (−17% reduction) and the cancer cost is increased slightly more than in the previous scenario, reaching 1.11% (an increase of 0.24% concerning the baseline). Thus, a 1% increase in cancer cost over the baseline leads to a reduction in the mortality rate of 1.4%. This means that an improvement in treatment helps the mortality reduction to occur more rapidly, compared to scenario 1, but the additional cost of its implementation could result in lower effectiveness per each unit of expenditure (i.e., there is a reduction in elasticity).

In the final Scenario 3, an improvement was added to the previous strategies to increase the effectiveness regarding stage I. This adjustment implies that the mortality rate in 2040 reaches 42.2, equivalent to a reduction of −38.3%, while the cost only increases by 0.2% to 0.76%. In terms of effectiveness, this implies that each 1% reduction in cost concerning the baseline generates a 4.1% reduction in mortality. Therefore, this strategy changes the relationship between expenditure and mortality, as expenditure can be reduced while mortality is also reduced at a high rate. In other words, efficiency improves substantially in both health and financial results.

On the other hand, a financial analysis was performed. The analysis concerning the baseline revealed that, while Scenarios 1 and 2 result in increased spending, Scenario 3 can be considered a long-term investment (Table 7). In Scenario 1, additional annual funding of 14% to 2030 and 15% to 2040 is required. In Scenario 2, these percentages increase to 20% and 23%, respectively. Meanwhile, in Scenario 3, an increase in spending is observed in the first years, but then there is a significant reduction in spending. This means that, by 2027, the internal rate of return (IRR) of the higher expenditure to the baseline is negative (−18.2%), indicating an investment stage. Then, the returns of the higher spending in the first years of this scenario increase, reflecting lower cancer spending and an IRR of 18.7% to 2040.

## 4. Discussion

In this work, we proposed and validated a mathematical model that can represent the actual dynamics within the cancer healthcare system in a reliable way, which is schematized in Figure 1. The developed model follows the guidelines proposed in [17] for discrete-time structures and, due to the compartmental nature of the agents and processes involved in cancer healthcare (see Figure 1), it can be separated into several autonomous components that can act as isolated models (see Figure 2, Figure 4, and Figure 5). We exploited this last feature to treat the flows of undiagnosed and diagnosed cancer patients as independent systems, whose parameters could be reliably estimated simply by making some natural assumptions, such as an almost steady state for the health system, which slowly changes due to population dynamics, and suggesting the fate of the patients to be almost decided over the first year of treatment (see Table 1 and Table 3). Finally, we used the models for age groups (see Figure 3) with calibrated parameters to test several hypotheses related to strategies and public policies with the potential to improve principal epidemiological indicators while reducing the long-term costs in the health system associated with cancer (see Figure 7 and Figure 9).

Due to limitations inherent to the available data, we were only able to model some of the processes of the cancer healthcare system related to the diagnosis, treatment, evolution, and prevention of the disease, ignoring its real complexity in terms of certain components of the health system. Such limitations led to the proposal of stronger assumptions; for instance, it is known that cancer is a highly heterogeneous disease [7], even at cellular level [45,46], which clearly runs against our Assumption 3. Although Assumption 7—related to infinite capacity of the health system—is more common in the literature (see, e.g., [47]), several works have remarked on the need to overcome it [2,48,49]. Finally, Assumption 6—related to independence of gender and age—seems to be wrong in light of the findings in [50,51], despite the fact that such a link remains unclear [52]. However, the availability of new data and models regarding the behavior of such factors will help to overcome these assumptions, possibly leading the proposed structure to become nonlinear. On the other hand, most of the parameters we identified regarding the costs, detection, progression, and recovery of the disease may remain useful for future work, due to the way in which we proposed the model (i.e., through the integration of components based on the natural history of the disease).

We chose Colombia as a case study as we believe it to be a strategic location for the development of long-term public policies aimed at improving the cancer healthcare process. Notably, Colombia currently has a relatively low proportion of the population over 65 years of age (8.5%, 63rd percentile worldwide), compared to countries with advanced aging (14% or more, 75th percentile). This is reflected in its relatively low expenditure on cancer as a proportion of GDP (0.37%), below the OECD average of 0.43% in 2009 [41]. However, it is higher than that of Denmark (0.25%) and similar to that of the United Kingdom (0.33%), countries with a population over 65 years of age between 18–20%. It is important to note that no health system in the world is entirely identical, as the idiosyncratic elements of each country strongly shape its health management structure.

Comparing the Colombian health system with characteristics reviewed in the case of Colombia for a group of countries from different continents and income levels (20 in total) by the Commonwealth Fund [53], the following observations can be made. In terms of universal coverage, Colombia shares this characteristic with most of the countries reviewed, except for India, China, and the United States. Regarding financing, the Colombian model is similar to that in countries such as England, Germany, India, Japan, and the United States. The Colombian health system relies on mandatory contributions from employers, employees, and the government. In terms of the benefits package, the Colombian health system includes a benefits package with a list of exclusions, similar to countries such as Australia, Brazil, Germany, and Singapore. Regarding access to private insurance, Colombia is comparable to most countries, as it provides access to private insurance, although the extent of access may vary among different countries. In terms of hospital supply, like many of the countries reviewed, Colombia has a mixed supply of hospitals. By examining these characteristics, we can gain insights into how the conclusions of this work on the Colombian health system can be extrapolated to other countries.

We think that the strategy followed in this paper to estimate model parameters was novel and successful as, unlike most of previous works we found in the field (e.g., [54,55,56]), we maintained a direct and natural meaning for the parameters while proposing structures whose outputs correspond—as much as possible—with variables typically measured by the health system, such as five-year survival percentages and the percentage of patients detected at each stage of the disease (see Table 2). The exception to this strategy was the total spending on cancer healthcare per level of treatment, as we inferred relevant values from the work of Gamboa et al. [30] and data available from the Colombian database SISPRO, assuming the first year of cancer treatment to be determinant for the patient survival, as detailed in Section 3.3.3. This last process, as well as the one for assigning different five-year survivals to each of the three levels for quality treatment identified, constitute the most ambitious and risky parts of this work.

We claimed the three different levels of quality treatment for Colombia to behave in a similar way in terms of the difference in survival indicators for some countries whose information was available at GLOBOCAN. In this way, we assumed that the worst-quality level in the Colombian health system has the same potential to improve as the worst-observed performance in the cluster of countries stated in Section 3.3.2 regarding the best-observed performance; that is, the Colombian health system has a variability in its quality similar to the variability in the performance of the cluster of countries. For further information, we refer the reader to GLOBOCAN [20]. Nevertheless, such strong assumptions regarding different treatment qualities and the relevance of the first year of treatment—which also leads to the highest spending in this year—seem to be appropriate, considering the validation results shown in Figure 6. It is necessary to point out that we did not perform any model fitting to the outputs chosen for validation, which means that the model would only adjust these curves in the case that it is a good representative of the cancer healthcare system, despite the assumptions that we made.

According to data stored in the IHME database [57], the monthly mortality rate per 100,000 population in Colombia presents an increasing trend, with a minimum value of 6.8 between the years 2015 and 2019. This estimated mortality rate significantly exceeds the official data reported by CAC [18], indicating a maximum value of 3.4 for the same time period, as shown in Figure 6 (left). However, the data from the CAC reveal a sudden increase from 2019 to 2021, surpassing a monthly mortality rate per 100,000 population of 5.5, which agrees with the model output for that period (see Figure 6 left). We hypothesize that this change in behavior, leading to a closer approximation of the IHME data by the official Colombian data, is related to a shift in the nature of data collection methods employed by the Colombian government. Hence, it is likely that the official data for the period 2015–2019 underestimated the actual mortality rate. Our model estimate represents an intermediate scenario between the official mortality rate and the one reported by the IHME for that period, demonstrating a good result by accounting for the discrepancy and uncertainty in the data from both sources.

A comparison of our results with those of Ward et al. [1] for Colombia also revealed significant disparities. While their baseline scenario estimated a total of 364,903 deaths over an 11-year period (2020–2030), our calculations (2022–2032) revealed a total of 441,861 deaths, for a 20% increase. However, one key difference is that, while they calibrated their indicator using GLOBOCAN data, we calibrated our indicator using a combination of data sources, including GLOBOCAN, Cancer Today, and national data.As a result, our data may be more accurate, as we accounted for the potential underestimation when using a single data source. On the other hand, we found the case of the costs projected in Figure 6 (left) to be similar to those projected by Mariotto et al. [3] over a 10-year horizon for both cases: 31% in our case versus 33% growth in spending relative to GDP.

It is worth noting that the scenarios analyzed in this study are not directly comparable to those of Ward et al. [1], as the variations and calibrations differ. While they tested various scenarios including improvements in imaging, treatment, quality, and combinations of these, with the scaling-up of conditions estimated by reaching the mean value of high-income countries, our study followed a different method (as described in Section 3.5.2). Additionally, they did not consider early detection in their scenario of improved imaging, which is the key element in our first scenario. However, when comparing our results for Scenarios 1 and 3 with their baseline projection, we found that while they reduced the number of deaths by 8%, our early detection scenario (which includes improvements in imaging) resulted in a 20% reduction. In their comprehensive treatment scenario, they obtained a 12% reduction, whereas our comprehensive scenario (No. 3) resulted in a 31% reduction in mortality. This highlights the importance of moving towards a comprehensive approach to treatments in order to achieve better health outcomes.

The SA results obtained for the model output T(t)—which represents the redistribution of costs due to the patients who died—allowed us to identify those key components in the structure of the cancer healthcare system. Table 5 summarizes the relevance of the parameters to the behavior of the model when we explored different combinations of their values. We chose to vary the value of each parameter by 50% with respect to their nominal value. In this way, we explored scenarios where the cancer healthcare system improved or deteriorated the performance of their components up to 50%. We excluded parameters related to treatment cost for the SA, as we already modeled inflation and GDP growth as components of the system. Further, a prevention parameter, η, was tested within an interval ranging from no reduction up to 30% of reduction in the cancer incidence. The transition speed parameter ($\beta_{1}$) and the abruptness of transition parameter ($\beta_{2}$) were tested with a nominal value of 0.3 for each, which was arbitrarily determined; however, this value can well represent a scenario of health policy reform followed by a transition period.

As expected, the prevention parameter η was the most relevant parameter in the model, although it was closely followed by the recovery probability under treatment 3 at stage IV ($\gamma [s_{4}T_{3}]$). It was observed that the detection parameters (δ) appeared below 2% of sensitivity contribution, which suggests that improving cancer screening may have a minor impact in the balance of efficiency in the cancer healthcare system (i.e., early cancer detection causes an increase in the number of patients in the system at early stages, but also leads to an increase in treatment spending). Indeed, we proved such behavior to be right under the assumptions of the model through testing Hypothesis 1 (see Figure 9) and by MC filtering (see Figure 8). Another result that drew our attention from the SA in Table 5 was the fact that parameters related to treatment quality for patients (λ) were below the 1% of sensitivity contribution, suggesting the existence of a very linear relationship between spending and survival, causing the model output (T(t)) to stay equal under variations in λ.

On the other hand, the behavior of the yellow lines regarding the behavior of red lines in the MC filtering results for sensitive model parameters (see Figure 8) indicated the complexity of trying to improve the current configuration of the cancer healthcare system. The red lines in the MC filtering results allowed us to identify the values of the parameters that led to improved efficiency (output T(t) below the red line in Figure 7), while the yellow lines are related to those values that decreased efficiency (output T(t) above the red line in Figure 7). Thus, we can see that a low prevention rate and lesser effectiveness of cancer treatment (γ,α) are directly linked to worse scenarios. However, there was no single parameter that was strongly linked to better scenarios, as the red lines displayed the same behavior as the blue lines in Figure 8. That is a relevant result, as it suggests that only improving several components in the current cancer healthcare system as a whole entity can lead to a better scenario. Furthermore, we tested this idea when defining Scenario 3 (see Figure 9) as the simultaneous combination of Hypotheses 1–3, which did not achieve good results individually (see Scenarios 1 and 2 in Figure 9). Although we did not model the delay between diagnosis and start of treatment, we hypothesize that it was included in the values for progression parameters α, which could explain, to some extent, the high values estimated for those parameters, including the fact that the progression speed was the same for diagnosed and undiagnosed patients. We encourage readers to refer to Appendix D for a more in-depth explanation regarding the impact of the high values for the progression parameters on the behavior of the model.

From the results of Scenarios 1–3, we infer that there exist two key strategies for improving cancer treatment conditions: early detection and higher efficiency in the early stages. Moreover, we found that moving to better treatments is a good way to improve health outcomes, even though the mortality reduction is smaller relative to the growth in costs. The different strategies evaluated achieved long-term reductions in the mortality rate; however, the best-performing strategy was derived under the comprehensive interventions (i.e., combining early detection with improvements in treatment efficiency). In this case, mortality went from a 9% reduction (with only improvements in early detection) to a 40% reduction over an 18-year horizon when changes in treatment efficiency were included. As for the financial sustainability of these changes, the results were not desirable when only early detection and/or treatment with high health improvement results were implemented, increasing the cost between 78% and 98% over the same horizon (i.e., additional annual funding requirements between 14% and 23% in each scenario, respectively). Meanwhile, in the best-case scenario, rates of return of 18.7% to 2040 were found to be achievable.

## 5. Conclusions

The results of this study provide important insights for the development of public health policy in Colombia. In alignment with the global focus on comprehensive cancer treatment, it is crucial for Colombia to prioritize a patient-centered approach in its cancer treatment efforts. Furthermore, it has been highlighted that prevention measures are essential in reducing the incidence of cancer and that cancer prevention policies in Colombia are currently inadequate, at least according to a review of the available databases and the operation of the health system. To address this problem, it is recommended that a prevention policy be integrated into the care policy and that clear responsibilities for actors in the health system be established. Additionally, it should be noted that coordination and stakeholder involvement is crucial, and that current payment mechanisms in Colombia do not align with the recommended approach. Therefore, it is suggested that more flexible regulatory frameworks be created to allow for contract models that promote comprehensiveness in cancer treatment and spending on cancer prevention.

Despite the limitations of our study, the results obtained are crucial for informing the debate on increasing cancer spending in healthcare systems, particularly in the context of Colombia. The literature in this field has emphasized various strategies for the improvement of cancer healthcare models, such as prevention. For instance, Vos et al. [58] systematically evaluated the effectiveness of 123 preventive interventions and 27 treatment interventions across a range of health issues in Australia, including 10 specifically related to cancer. Among these, they found that six were cost-effective while four were not. Additionally, many interventions with high cost-effectiveness, such as tobacco and alcohol taxes, also have a significant impact on the development of oncological diseases. Furthermore, interventions that promote early detection have been shown to have positive results. For example, Laudicella et al. [4] have reported significant savings due to earlier detection in a patient-level study conducted in England.

It is essential to focus interventions on innovations or treatments that have demonstrated significant health outcomes, rather than including new technologies with little evidence and high costs into healthcare plans. The Lancet Oncology Commission [59] has compiled evidence and outlined key measures in this regard, including the promotion of low-cost innovations, the increased use of off-patent products, more research on comorbidities, and reducing the number of technologies that only provide marginal benefits. Additionally, reducing bureaucracy for cancer issues and promoting more comprehensive research to improve evidence-based policies are important measures. Better practices among oncology physicians are crucial for implementing changes and reducing costs through their practice. For instance, Smith and Hillner [60] have provided recommendations for improving the behavior of these physicians.

Comprehensiveness of treatment is a key measure to reduce mortality and costs, as it increases coordination between actors and ensures timeliness. The results of this study support the implementation of such measures. Furthermore, the implementation of these measures necessitates the development of appropriate regulatory frameworks that facilitate the adoption of contracts that incentivize actors to achieve improved health and financial outcomes. The literature in this field has identified various forms of contracting—such as bundled or packaged payment models [61,62], pay-for-performance [63,64], and Accountable Care Organizations [61,65]—that can provide positive incentives for actors to achieve better health and financial results.

Finally, considering the availability of data on the functioning of cancer healthcare systems in most countries and the common underlying dynamics of the cancer cycle (as depicted in Figure 1), we believe that the proposed methodology can be employed and adapted for numerous other case studies with only minor modifications. Furthermore, we recognize that necessary adjustments would be required for components of the systems that we were unable to model in this study. Thus, we contend that the present work and its findings hold a general perspective that is applicable to most cancer healthcare systems, while acknowledging that specific parameter combinations must be estimated for each unique case. We consider this work as a foundational framework for proposing and validating more realistic and accurate models, aimed at simulating the effects of public policies and their impacts on the various components of the cancer healthcare system.

## Figures and Tables

**Figure 1 ijerph-20-06740-f001:**
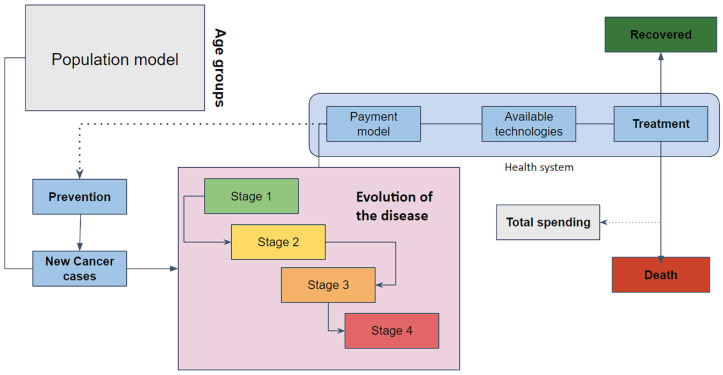
Scheme of the cancer cycle before and after patients enter the Colombian cancer healthcare system. Bold letters indicate those components of the system for which sufficient data were found to model them with reliable parameters. Dotted lines represent indirect interactions.

**Figure 3 ijerph-20-06740-f003:**
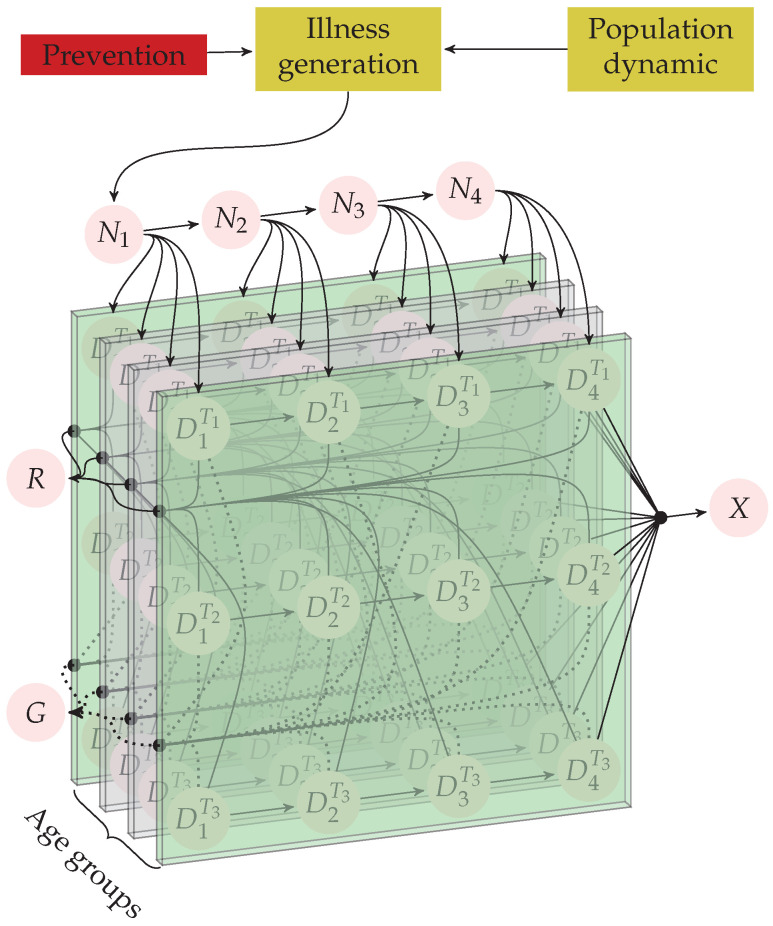
Representation of the proposed mathematical model from the perspective of meta-population modeling to take age groups into account. Every model inside a panel is a copy of the model depicted in Figure 2.

**Figure 4 ijerph-20-06740-f004:**
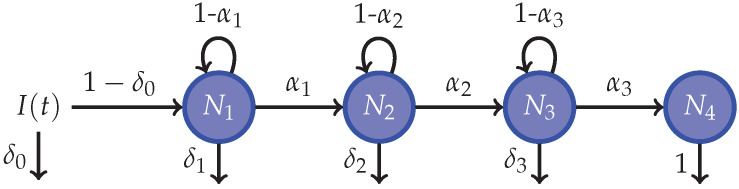
Auxiliary model for estimating parameters related to the undiagnosed states. This model represents the undiagnosed patient component in Figure 2 by explicitly including the parameters that determine the flows. Due to the prioritization of detection (δ) over progression (α), the flow from state $N_{1}$ to $N_{2}$ takes the form $\left(1-\delta_{1}\right)\alpha_{1}$, while the loop from $N_{1}$ to itself has the form $\left(1-\delta_{1}\right)\left(1-\alpha_{1}\right)$; the flow from state $N_{2}$ to $N_{3}$ has the form $\left(1-\delta_{2}\right)\alpha_{2}$, and the loop from $N_{2}$ to itself has the form $\left(1-\delta_{1}\right)\left(1-\alpha_{1}\right)$; and so on. We simplify the representation of the model by omitting this level of detail to enhance the interpretability of the figure. For a more comprehensive understanding, please refer to Section 3.2.1.

**Figure 5 ijerph-20-06740-f005:**
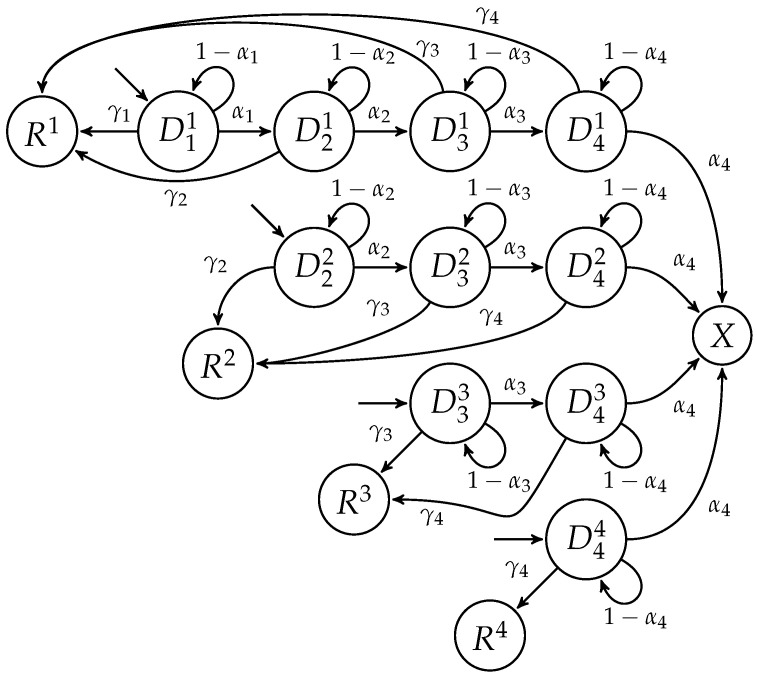
Auxiliary model for estimating parameters related to the diagnosis state (see Section 3.3.2). This model represents the component of diagnosed patients in Figure 2 for a single level of treatment quality, while tracing the flow of patients according to the stage of the disease at the time of diagnosis. The superindices for each state indicate the stage at the time of diagnosis. *X* represents the individuals who pass away at each time step. Similar to the diagram in Figure 4, the recovery process (γ) takes precedence over the transition process (α). Thus, the flow from state $D_{1}^{1}$ to $D_{2}^{1}$ takes the form $\left(1-\gamma_{1}\right)\alpha_{1}$, and the loop from state $D_{1}^{1}$ to itself has the form $\left(1-\gamma_{1}\right)\left(1-\alpha_{1}\right)$. However, we omit this level of detail to enhance interpretability. The unlabeled arrows at the beginning of each chain of states with the same superindex represent the initial condition of diagnosed patients.

**Figure 6 ijerph-20-06740-f006:**
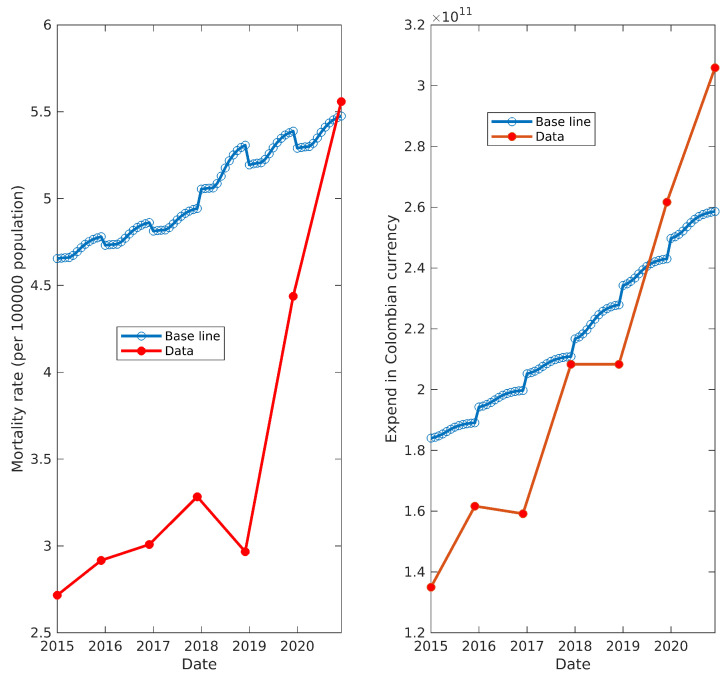
Validation of the baseline behavior of the proposed estimated model. (Left) Mortality rate per 100,000 population (shown in blue) compared to the non-fitted real data (shown in red). The blue line represents the model output with nominal parameter values from Table 1, Table 3 and Table 4, which serves as the baseline for further comparisons with scenarios involving different parameter values. (Right) Expenditure in Colombian currency for cancer care (shown in blue) compared to the non-fitted real data (shown in red). The blue line represents the model output with nominal parameter values from Table 1, Table 3 and Table 4, which serves as the baseline for further comparisons with scenarios involving different parameter values.

**Figure 7 ijerph-20-06740-f007:**
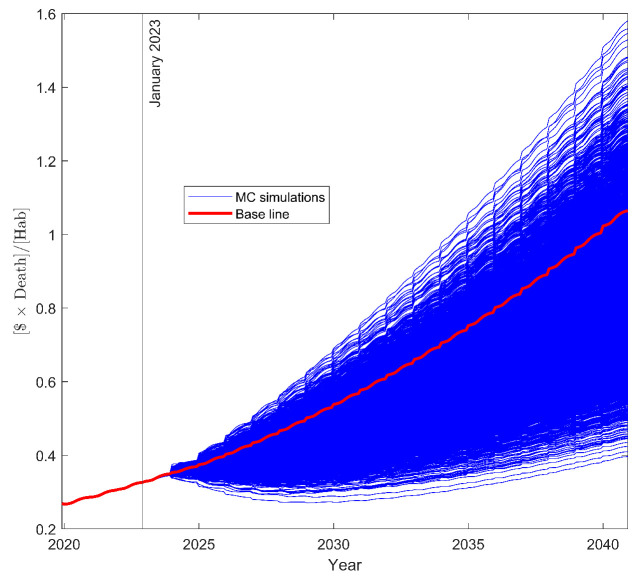
Uncertainty analysis for target model output *T* using intervals in Table 5 for those parameters with $\%SST_{i}$ greater than 1%. The remaining parameters were kept fixed at their nominal values.

**Figure 8 ijerph-20-06740-f008:**
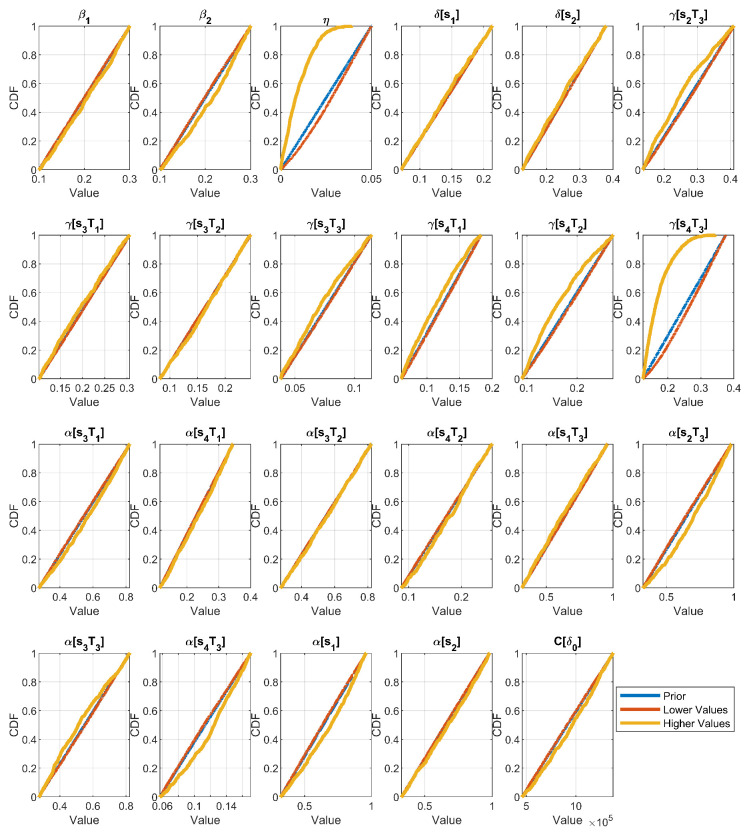
Results of the MC filtering applied to UA in Figure 7. The difference between eCDF for lower ($\theta_{i}^{0}$) and higher ($\theta_{i}^{1}$) values was greater for those parameters with large $\%SST_{i}$ values (see Table 5). The exception to this rule was the parameter $\alpha [s_{1}]$.

**Figure 9 ijerph-20-06740-f009:**
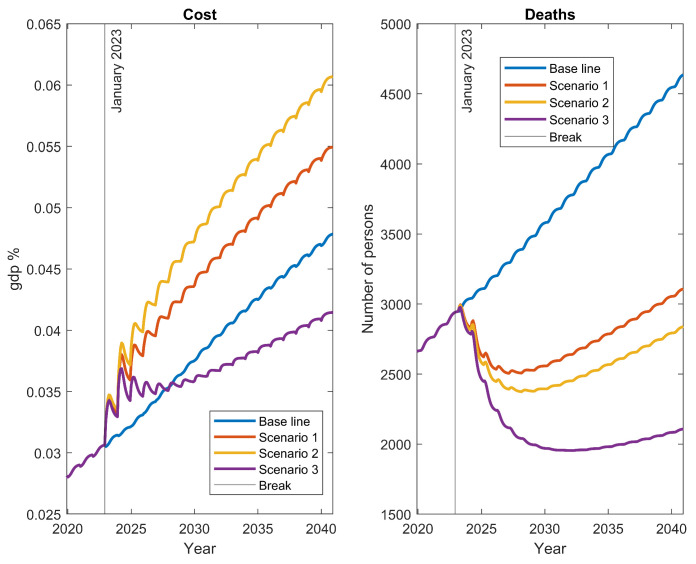
Simulation of the three different scenarios induced by Hypotheses 1–3. Scenario 3 is the only one that leads to a long-term reduction in cancer spending.

**Table 1 ijerph-20-06740-t001:** Parameters estimated using the fitting auxiliary model in Figure 4 to real data following the proposed methodology. We decided to make the stage to which each parameter is related explicit using the [·] notation. As all of the parameters are probabilities; the estimation range was their whole natural domain.

Parameter	Meaning	Estimation Range	Estimated Interval	Nominal Value
$\delta [s_{0}]$	Early detection prob.	[0, 1]	[0.0875, 0.0875]	0.0875
$\delta [s_{1}]$	Stage I detection prob.	[0, 1]	[0.1315, 0.1538]	0.1422
$\delta [s_{2}]$	Stage II detection prob.	[0, 1]	[0.2169, 0.2587]	0.2525
$\delta [s_{3}]$	Stage III detection prob.	[0, 1]	[0.3175, 0.3852]	0.3489
$\alpha [s_{1}]$	Stage I progress prob.	[0, 1]	[0.5849, 0.7017]	0.6398
$\alpha [s_{2}]$	Stage II progress prob.	[0, 1]	[0.5357, 0.6749]	0.6532
$\alpha [s_{3}]$	Stage III progress prob.	[0, 1]	[0.4756, 0.6405]	0.5479

**Table 2 ijerph-20-06740-t002:** Summary of information available about five-year survival for cancer in Colombia. Data in each Stage column represent the percentage of individuals who surpassed the five-year survival threshold from a starting population given by the number of patients in the Cases column. Using the number of cases as weights, we estimated the general expected cancer five-year survival in the Weighted mean row. Additionally, from such information and the variation in survival in the GLOBOCAN database, we estimated extreme scenarios for general survival.

Type	Stage 1	Stage 2	Stage 3	Stage 4	Cases
Breast	0.902	0.885	0.789	0.676	24,460
Prostate	0.828	0.783	0.725	0.618	13,190
Colon & rectal	0.672	0.665	0.563	0.443	9330
Stomach	0.458	0.44	0.36	0.214	4218
Lung	0.35	0.249	0.231	0.088	2027
Weighted mean	0.787	0.762	0.678	0.562	
Extreme scenarios	[0.707, 0.867]	[0.662, 0.861]	[0.576, 0.780]	[0.376, 0.747]	

**Table 3 ijerph-20-06740-t003:** Parameters estimated from fitting auxiliary model in Figure 5 to real data following the proposed methodology. We decided to make the stage and the treatment level to which each parameter is related explicit using the [·] notation. As all the parameters are probabilities, the estimation range for most of them was their whole natural domain. For the case of stage progression, we set the nominal values for progression in Table 1 as superior bounds for their estimation range.

Parameter	Meaning	Estimation Range	Estimated Interval	Nominal Value
$\gamma [s_{1}T_{1}]$	Recovery stage I, treatment 1 prob.	[0, 1]	[0.088, 0.088]	0.088
$\gamma [s_{1}T_{2}]$	Recovery stage I, treatment 2 prob.	[0, 1]	[0.07, 0.07]	0.07
$\gamma [s_{1}T_{3}]$	Recovery stage I, treatment 3 prob.	[0, 1]	[0.031, 0.031]	0.031
$\gamma [s_{2}T_{1}]$	Recovery stage II, treatment 1 prob.	[0, 1]	[0.141, 0.141]	0.141
$\gamma [s_{2}T_{2}]$	Recovery stage II, treatment 2 prob.	[0, 1]	[0.186, 0.186]	0.186
$\gamma [s_{2}T_{3}]$	Recovery stage II, treatment 3 prob.	[0, 1]	[0.273, 0.273]	0.273
$\gamma [s_{3}T_{1}]$	Recovery stage III, treatment 1 prob.	[0, 1]	[0.204, 0.204]	0.204
$\gamma [s_{3}T_{2}]$	Recovery stage III, treatment 2 prob.	[0, 1]	[0.164, 0.164]	0.164
$\gamma [s_{3}T_{3}]$	Recovery stage III, treatment 3 prob.	[0, 1]	[0.076, 0.076]	0.076
$\gamma [s_{4}T_{1}]$	Recovery stage IV, treatment 1 prob.	[0, 1]	[0.122, 0.122]	0.122
$\gamma [s_{4}T_{2}]$	Recovery stage IV, treatment 2 prob.	[0, 1]	[0.181, 0.181]	0.181
$\gamma [s_{4}T_{3}]$	Recovery stage IV, treatment 3 prob.	[0, 1]	[0.250, 0.250]	0.25
$\alpha [s_{1}T_{1}]$	Progression from stage I to stage II, treatment 1.	[0, 0.640]	[0.640, 0.640]	0.64
$\alpha [s_{1}T_{2}]$	Progression from stage I to stage II, treatment 2.	[0, 0.640]	[0.640, 0.640]	0.64
$\alpha [s_{1}T_{3}]$	Progression from stage I to stage II, treatment 3.	[0, 0.640]	[0.640, 0.640]	0.64
$\alpha [s_{2}T_{1}]$	Progression from stage II to stage III, treatment 1.	[0, 0.653]	[0.653, 0.653]	0.653
$\alpha [s_{2}T_{2}]$	Progression from stage II to stage III, treatment 2.	[0, 0.653]	[0.653, 0.653]	0.653
$\alpha [s_{2}T_{3}]$	Progression from stage II to stage III, treatment 3.	[0, 0.653]	[0.653, 0.653]	0.653
$\alpha [s_{3}T_{1}]$	Progression from stage III to stage IV, treatment 1.	[0, 0.548]	[0.548, 0.548]	0.548
$\alpha [s_{3}T_{2}]$	Progression from stage III to stage IV, treatment 2.	[0, 0.548]	[0.548, 0.548]	0.548
$\alpha [s_{3}T_{3}]$	Progression from stage III to stage IV, treatment 3.	[0, 0.548]	[0.548, 0.548]	0.548
$\alpha [s_{4}T_{1}]$	Death stage IV, treatment 1 prob.	[0, 1]	[0.230, 0.230]	0.23
$\alpha [s_{4}T_{2}]$	Death stage IV, treatment 2 prob.	[0, 1]	[0.172, 0.172]	0.172
$\alpha [s_{4}T_{3}]$	Death stage IV, treatment 3 prob.	[0, 1]	[0.114, 0.114]	0.114

**Table 4 ijerph-20-06740-t004:** Costs estimated from real data for the three different levels of breast cancer treatment for the year 2012 (in Colombian currency). To extrapolate those costs for all types of cancer in every year, we considered the inflation rate for Colombia reported in [42] and applied the correction factor given in Equation (Equation 10) to each value.

Parameter	Description	Nominal
$C[s_{1}T_{1}]$	Monthly average cost of patient at stage I under treatment 1	747,500.000
$C[s_{1}T_{2}]$	Monthly average cost of patient at stage I under treatment 2	782,500.000
$C[s_{1}T_{3}]$	Monthly average cost of patient at stage I under treatment 3	1,008,333.333
$C[s_{2}T_{1}]$	Monthly average cost of patient at stage II under treatment 1	3,966,666.667
$C[s_{2}T_{2}]$	Monthly average cost of patient at stage II under treatment 2	4,325,000.000
$C[s_{2}T_{3}]$	Monthly average cost of patient at stage II under treatment 3	6,783,333.333
$C[s_{3}T_{1}]$	Monthly average cost of patient at stage III under treatment 1	4,666,666.667
$C[s_{3}T_{2}]$	Monthly average cost of patient at stage III under treatment 2	5,325,000.000
$C[s_{3}T_{3}]$	Monthly average cost of patient at stage III under treatment 3	8,666,666.667
$C[s_{4}T_{1}]$	Monthly average cost of patient at stage IV under treatment 1	9,083,333.333
$C[s_{4}T_{2}]$	Monthly average cost of patient at stage IV under treatment 2	12,000,000.000
$C[s_{4}T_{3}]$	Monthly average cost of patient at stage IV under treatment 3	12,416,666.667
C[δ]	Average costs of diagnosis per diagnosed patient	916,000.000

**Table 5 ijerph-20-06740-t005:** Summary of total-order sensitivity indices for a variation of 50% in the nominal values of the parameters.

Parameter	Description	Range	Nominal	%SSTi
η	Prevention parameter	[0.00, 0.050]	0.00	19.07%
$\gamma [s_{4}T_{3}]$	Recover [stage IV, treatment 3] prob.	[0.125, 0.380]	0.25	18.52%
$\alpha [s_{4}T_{3}]$	Death [stage IV, treatment 3] prob.	[0.057, 0.170]	0.11	4.69%
$\gamma [s_{4}T_{2}]$	Recover [stage IV, treatment 2] prob.	[0.091, 0.270]	0.18	4.64%
$\gamma [s_{2}T_{3}]$	Recover [stage II, treatment 3] prob.	[0.137, 0.410]	0.27	4.27%
$\beta_{2}$	Transition speed parameter	[0.100, 0.300]	0.20	3.83%
$\gamma [s_{4}T_{1}]$	Recover [stage IV, treatment 1] prob.	[0.061, 0.183]	0.12	3.44%
$\alpha [s_{2}T_{3}]$	Progression [stage II, treatment 3] prob.	[0.327, 0.980]	0.65	3.23%
$\alpha [s_{3}T_{3}]$	Progression [stage III, treatment 3] prob.	[0.274, 0.822]	0.55	3.03%
$\alpha [s_{4}T_{2}]$	Death [stage IV, treatment 2] prob.	[0.086, 0.259]	0.17	2.66%
$\beta_{1}$	Abruptness of transition parameter	[0.100, 0.300]	0.20	2.37%
$\gamma [s_{3}T_{3}]$	Recover [stage III, treatment 3] prob.	[0.038, 0.114]	0.08	2.33%
$\alpha [s_{4}T_{1}]$	Death [stage IV, treatment 1] prob.	[0.115, 0.344]	0.23	2.23%
$\alpha [s_{3}T_{1}]$	Progression [stage IV, treatment 3] prob.	[0.274, 0.822]	0.55	2.02%
$\gamma [s_{3}T_{1}]$	Recover [stage III, treatment 1] prob.	[0.102, 0.306]	0.20	1.85%
$\alpha [s_{1}]$	Undiagnosed progression at stage I prob.	[0.320, 0.960]	0.64	1.58%
$\delta [s_{1}]$	Stage I detection prob.	[0.071, 0.213]	0.14	1.56%
$\delta [s_{2}]$	Stage II detection prob.	[0.126, 0.379]	0.25	1.46%
$\gamma [s_{3}T_{2}]$	Recover [stage III, treatment 2] prob.	[0.082, 0.246]	0.16	1.43%
$\alpha [s_{1}T_{3}]$	Progression [stage I, treatment 3] prob.	[0.320, 0.960]	0.64	1.35%
$\alpha [s_{2}]$	Undiagnosed progression at stage II prob.	[0.327, 0.980]	0.65	1.21%
C[δ]	Average costs of diagnosis per patient	[458,000, 1,374,000]	916,000.00	1.09%
$\alpha [s_{3}T_{2}]$	Progression [stage III, treatment 2] prob.	[0.274, 0.822]	0.55	0.93%
$\lambda [s_{2}T_{2}]$	Prob. of treatment 2 at stage II	[0.130, 0.390]	0.26	0.90%
$\delta [s_{3}]$	Stage III detection prob.	[0.174, 0.523]	0.35	0.84%
$\lambda [s_{1}T_{1}]$	Prob. of treatment 1 at stage 1	[0.098, 0.294]	0.20	0.84%
$\gamma [s_{2}T_{2}]$	Recover [stage II, treatment 2] prob.	[0.093, 0.279]	0.19	0.84%
$\alpha [s_{2}T_{1}]$	Progression [stage II, treatment 1] prob.	[0.327, 0.980]	0.65	0.84%
$\lambda [s_{1}T_{2}]$	Prob. of treatment 2 at stage I	[0.130, 0.390]	0.26	0.81%
$\lambda [s_{3}T_{2}]$	Prob. of treatment 2 at stage III	[0.130, 0.390]	0.26	0.79%
$\lambda [s_{2}T_{1}]$	Prob. of treatment 1 at stage II	[0.098, 0.294]	0.20	0.70%
$\lambda [s_{4}T_{1}]$	Prob. of treatment 1 at stage IV	[0.098, 0.294]	0.20	0.67%
$\alpha [s_{3}]$	Undiagnosed progression at stage III prob.	[0.274, 0.822]	0.55	0.64%
$\delta [s_{0}]$	Early detection prob.	[0.044, 0.131]	0.09	0.63%
$\gamma [s_{2}T_{1}]$	Recover [stage II, treatment 1] prob.	[0.071, 0.212]	0.14	0.57%
$\lambda [s_{4}T_{2}]$	Prob. of treatment 2 at stage IV	[0.130, 0.390]	0.26	0.52%
$\alpha [s_{2}T_{2}]$	Progression [stage II, treatment 2] prob.	[0.327, 0.980]	0.65	0.47%
$\lambda [s_{3}T_{1}]$	Prob. of treatment 1 at stage III	[0.098, 0.294]	0.20	0.42%
$\alpha [s_{1}T_{1}]$	Progression [stage I, treatment 1] prob.	[0.320, 0.960]	0.64	0.24%
$\gamma [s_{1}T_{3}]$	Recover [stage I, treatment 3] prob.	[0.016, 0.047]	0.03	0.20%
$\gamma [s_{1}T_{2}]$	Recover [stage I, treatment 2] prob.	[0.035, 0.105]	0.07	0.11%
$\gamma [s_{1}T_{1}]$	Recover [stage I, treatment 1] prob.	[0.044, 0.132]	0.09	0.11%
$\alpha [s_{1}T_{2}]$	Progression [stage I, treatment 2] prob.	[0.320, 0.960]	0.64	0.07%

**Table 6 ijerph-20-06740-t006:** Impact of policy scenarios on health and economic outcomes.

Scenario	Result	Indicator	Anual			% Change to 2020	
			**2022**	**2030**	**2040**	**2030**	**2040**
Baseline	Cancer cost	% GDP real	0.37%	0.46%	0.57%	24.4%	54.9%
		Per capita (USD 2015)	USD 24.3	USD 36.0	USD 57.4	48%	137%
	Mortality	Total	35,291	43,462	55,026	23%	56%
		Per 100,000 inhabitants	68.4	78.1	92.4	14%	35%
Scenario 1	Cancer cost	% GDP real	0.37%	0.53%	0.66%	45.2%	78.2%
		Per capita (USD 2015)	USD 24.3	USD 42.1	USD 66.1	73%	172%
	Mortality	Total	35,291	30,928	36,950	−12%	5%
		Per 100,000 inhabitants	68.4	55.5	62.0	−19%	−9%
Scenario 2	Cancer cost	% GDP real	0.37%	0.58%	0.72%	57.8%	96.9%
		Per capita (USD 2015)	USD 24.3	USD 45.7	USD 73.0	88%	201%
	Mortality	Total	35,291	28,890	33,767	−18%	−4%
		Per 100,000 inhabitants	68.4	51.9	56.7	−24%	−17%
Scenario 3	Cancer cost	% GDP real	0.37%	0.43%	0.50%	18%	35%
		Per capita (USD 2015)	USD 24.3	USD 34.2	USD 50.0	41%	106%
	Mortality	Total	35,291	23,571	25,134	−33%	−29%
		Per 100,000 inhabitants	68.4	42.3	42.2	−38%	−38%

**Table 7 ijerph-20-06740-t007:** Financial analysis of scenarios.

Scenario	Indicator	Value
No. 1	Annual funding (%BL) to 2030	13.8%
	Annual funding (%BL) to 2040	14.7%
	Annual average funding (%GDP) to 2030	−0.07%
	Annual average funding (%GDP) to 2040	−0.07%
No. 2	Annual funding % to BL—2030	19.5%
	Annual funding % to BL—2040	23.0%
	Annual average funding (%GDP) to 2030	−0.09%
	Annual average funding (%GDP) to 2040	−0.12%
No. 3	Internal Rate of Return (IRR) to 2030	−18.2%
	Internal Rate of Return (IRR) to 2040	18.7%
	Annual average funding (%GDP), 2023–2027	−0.03%
	Average annual return to GDP (%GDP), 2028–2040	0.04%

## Data Availability

All our results can be replicated by running codes and using data stored at https://github.com/drojasd/MDPICancerPolicies (accessed on 27 February 2023). Data for Colombian population dynamics can be downloaded from https://www.dane.gov.co/index.php/estadisticas-por-tema/demografia-y-poblacion/proyecciones-de-poblacion (accessed on 10 October 2022). Data for the cluster of countries at GLOBOCAN can be found at https://gco.iarc.fr/survival/survmark/ (accessed on 12 October 2022).

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
