# Peer review of "Mathematical Modeling for the Assessment of Public Policies in the Cancer Health-Care System Implemented for the Colombian Case"

_ijerph, 2023, doi:10.3390/ijerph20186740_

Round 1

Reviewer 1 Report

Dear Authors,   

  

I read your mathematical modeling study of public policies in the cancer health-care system in Columbia with great interest and would like to congratulate the authors on an excellent and complex modelling case study.  

Please find attached my comments in the separate word document.  

Overall, I enjoyed reading the article but it was not entirely clear to me what the tested public policies were. Instead, it seems as if outcome scenarios were tested. Further clarifications and dicussions of generalisability would be much appreciated.

All the best for your future work.

Kind regards,

Reviewer 1

Author Response

Dear Reviewer,

Thank you for your positive feedback and your interest in our mathematical modeling study on public policies in the cancer health-care system in Colombia. We appreciate your congratulations and recognition of the complexity of our case study.

We would also like to express our gratitude for your valuable comments, which have helped us improve the clarity and discussion of our work. We understand your concern regarding the clarity of the tested public policies in our study. Upon careful consideration, we realized that our description of the tested policies could have been more explicit. We have now revised the manuscript to provide a clearer explanation of the specific policies that were tested in our modeling framework.

Furthermore, we have taken your suggestion to heart and have included additional discussions on the generalizability of our findings. We acknowledge that generalizability is an important aspect of any modeling study, and we have expanded our discussion to address this point more comprehensively. We now provide a thorough explanation of how our methodology and results can be extrapolated to other contexts, as well as the necessary modifications required for different health-care systems.

Once again, we sincerely appreciate your valuable feedback and suggestions, as they have significantly contributed to the improvement of our paper. We have carefully reviewed and considered each point in the separate Word document you provided, and we will now proceed to respond to each suggestion in detail.

Thank you for your time and support. Please find our point by point response to your comments in the attachment.

Best regards,

Reviewer 2 Report

This is an interesting analysis that may support decision making in Colombia. The approach is also interesting for other countries to set priorities in cancer prevention and treatment. A main problem of the paper is that it is very detailed and hard to read.

 I have a couple of problems with the paper in its current form:

1.      The scenarios are not well explained and introduced for the first time on page 22. However, the authors already refer to the scenarios on page 6. The presentation of the scenario analyses (including Figure 9 and Table 6) is a central message of the paper and therefore I think that these different scenarios should already be described in the introduction and methods section. And also in more accessible wording.

2.      The paper is too technical. This is a big problem for policy makers who are interested in setting healthcare priorities. It is not clear to me whether this paper is mainly a methodological paper or whether this paper aims to evaluates different innovation opportunities in cancer prevention and treatment. I think it is both, but I think the second approach is more interesting for the readers of this journal. I suggest to postpone most of the mathematics to the appendix and to draft this paper with a focus on Colombian healthcare priority setting.

 3.      The three scenarios should be explained in more detail. I also think that scenario 3 is not very realistic. In scenario 3, the probability of recovery (stage 1, treatment 3) is increased from .03 to 0.6,  which is a 20 fold increase in the treatment success rate. Indeed, this will eventually lead to lower costs and more benefit, but it seems a very hypothetical situation to me. Could the authors motivate the choice of the parameter values in their scenarios? A more detailed scenario analysis in which the treatment parameters take on multiple values would be informative.

 There are numerous potential inconsistencies/unclear sentences in the paper which should be explained better. These include:

 1.      P.6. l.232. theta-hat -> the hat symbol is used when an estimator of theta is presented. Is that the case here as well?

2.      P.6.l.242. CDF. I do not understand what is done at point 3. Please explain in appendix and in more detail.

3.      P.9.Figure 2. It seems that the worst quality treatment is offered first and the best quality treatment is offered last, because one moves from T1 to T2 to T3 in the figure. This seems very strange. What do the authors mean with worst and best quality treatment (see page 10)?  

4.      P12.Figure 4 and P14.Figure 5. Probabilities in the figures do not sum to one:

alpha + (1-alpha) + delta1 > 1

5.      Table 3. T3 is the best quality treatment, but the recovery probability after treatment T3 (gamma[s1T3] = 0.03) is smaller than the recovery probability after treatment T1 (gamma[s1T1] = 0.09).

6.      Figure 6. The fit of the model to the mortality data does not seem very good before 2020, so I do not fully agree with the authors that the fit of the model to the data is good. However, the fit is not that bad either. And why did the observed mortality increase sharply in 2020? Is it related to the pandemic?

Author Response

We appreciate the reviewer's recognition of the potential value of our analysis in supporting decision making in Colombia and other countries for setting priorities in cancer prevention and treatment. We agree that the paper contains detailed information and technical aspects that may make it challenging to read, especially for a broader audience that includes policy makers.

In response to this feedback, we have made significant efforts to improve the readability and accessibility of the paper. We have revised and restructured sections to ensure a clearer flow of information and have used more accessible language to explain complex concepts. Furthermore, we have provided additional context and explanations to help readers, including policy makers, better understand the implications and applicability of our approach in the healthcare priority-setting context.

While it is important to maintain the necessary level of detail for a rigorous analysis, we have carefully balanced it with the need for clarity and comprehension. Our aim is to ensure that the paper remains valuable to both technical experts and policy makers seeking to utilize our findings.

We hope that these revisions address the reviewer's concerns and improve the overall readability of the paper. We are grateful for the valuable feedback provided, which has enabled us to enhance the presentation of our work and make it more accessible to a wider audience.

Thank you once again for your time and valuable input. Please find our point-by-point response to your comments in the attachment.

Round 2

Reviewer 2 Report

The paper has substantially improved in clarity.